# EGGS: Exchangeable 2D/3D Gaussian Splatting for Geometry-Appearance Balanced Novel View Synthesis

Yancheng Zhang , Guangyu Sun , and Chen Chen

Institute of Artificial Intelligence, University of Central Florida
{yczhang, guangyu.sun, chen.chen}@ucf.edu

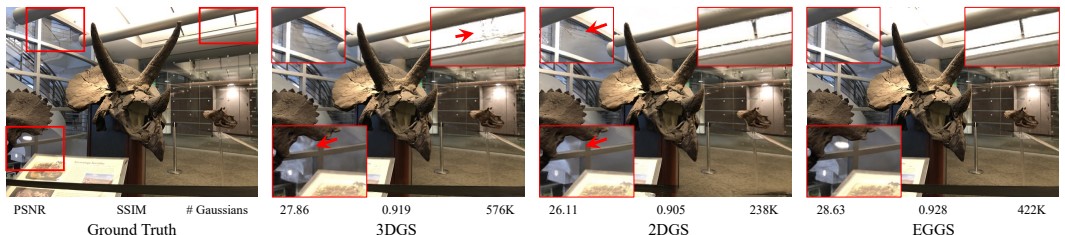

| PSNR | SSIM | # Gaussians | 27.86 | 0.919 | 576K | 26.11 | 0.905 | 238K | 28.63 | 0.928 | 422K |
| Ground Truth | | | | 3DGS | | | 2DGS | | | EGGS | |

Figure 1: Comparison of 3DGS, 2DGS, and our EGGS. While 3DGS achieves high-fidelity appearance, it often produces inaccurate geometry, with imprecise surfaces and blurred edges. 2DGS improves geometric consistency across views but suffers from reduced appearance quality due to over-smoothed surfaces and loss of detail. In contrast, EGGS employs an exchangeable hybrid Gaussian representation that achieves both accurate geometry and high-quality appearance.

## Abstract

Novel view synthesis (NVS) is crucial in computer vision and graphics, with wide applications in AR, VR, and autonomous driving. While 3D Gaussian Splatting (3DGS) enables real-time rendering with high appearance fidelity, it suffers from multi-view inconsistencies, limiting geometric accuracy. In contrast, 2D Gaussian Splatting (2DGS) enforces multi-view consistency but compromises texture details. To address these limitations, we propose Exchangeable Gaussian Splatting (EGGS), a hybrid representation that integrates 2D and 3D Gaussians to balance appearance and geometry. To achieve this, we introduce Hybrid Gaussian Rasterization for unified rendering, Adaptive Type Exchange for dynamic adaptation between 2D and 3D Gaussians, and Frequency-Decoupled Optimization that effectively exploits the strengths of each type of Gaussian representation. Our CUDA-accelerated implementation ensures efficient training and inference. Extensive experiments demonstrate that EGGS outperforms existing methods in rendering quality, geometric accuracy, and efficiency, providing a practical solution for high-quality NVS. Code and demo available at https://github.com/Fobow/EGGS.

## 1 Introduction

Novel view synthesis (NVS) is a fundamental task in computer graphics and computer vision, with broad applications in augmented reality (AR), virtual reality (VR), and autonomous driving [1, 2, 3]. Neural Radiance Fields (NeRF) [4] reconstruct implicit radiance fields via differentiable volume rendering. Despite achieving photorealistic appearance and accurate geometry, NeRF-based methods [5, 6, 7, 8, 9, 10] typically suffer from long training times and slow rendering speeds. 3D Gaussian Splatting (3DGS) [11] has emerged as an efficient alternative, leveraging anisotropic

39th Conference on Neural Information Processing Systems (NeurIPS 2025).

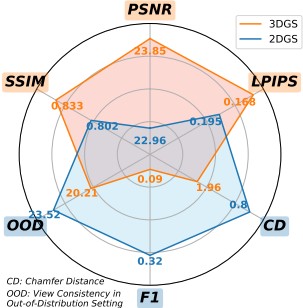

| Method | Gaussian Type | Raster-izer | Type Exchange | Regular-ization | Setting |
|---|---|---|---|---|---|
| **3DGS** SIGGRAPH'23 [11] | 3D | 3D | ✗ | - | General |
| **SuGaR** CVPR'24 [14] | 3D | 3D | ✗ | Normal | General |
| **GaussianPro** ICML'24 [13] | 3D | 3D | ✗ | Normal | General |
| **2DGS** SIGGRAPH'24 [12] | 2D | 2D | ✗ | Depth & Normal | General |
| **GS Surfels** SIGGRAPH'24 [15] | 2D | 3D⋆ | ✗ | Depth & Normal | General |
| **TextureGS** ECCV'24 [16] | 2D | 2D | ✗ | Depth & Normal | General |
| **HybridGS** CVPR'25 [17] | 3D + 2D* | 3D | ✗ | - | Transient |
| **HorizonGS** CVPR'25 [18] | 3D / 2D | 3D / 2D | ✗ | Depth & Normal | Varying-altitude |
| **Ours** | 3D + 2D | Hybrid | ✓ | Frequency | General |

Figure 2: Left: Comparison of 3DGS and 2DGS in **appearance** and **geometry** metrics. Right: Comparison between EGGS and related works. Prior works either use only single representation or do not explore complementary advantages of 3D and 2D Gaussians. ⋆ Gaussian Surfel [15] directly sets the $z$-scale of 3D Gaussian to zero and uses the rasterizer from 3DGS. * HybridGS [17] uses image-frame single-view 2D Gaussians [19, 20] instead of 2D Gaussians in the 3D space [12].

3D Gaussians for real-time, high-quality rendering. While 3DGS excels in appearance fidelity, its anisotropic nature often leads to multi-view inconsistencies, limiting geometric accuracy [12, 13]. As shown in Figure 1, this can lead to inaccurate edges and surfaces.

Following 3DGS, a line of work has focused on improving its geometric accuracy and reconstruction quality through additional regularization and novel representations, as shown in Figure 2 (right). SUGAR [14] and GaussianPro [13] introduce normal-based regularization, such as planar loss, to align Gaussian normals and encourage flatter shapes, thereby improving surface consistency. Gaussian Surfles [15] and GOF [21] incorporate additional geometry-aware constraints to enhance spatial coherence. 2D Gaussian Splatting (2DGS)[12] replaces 3D ellipsoids with 2D surfels, significantly improving multi-view consistency and geometric accuracy, as shown in Figure 2 (left). However, this comes at the cost of degraded appearance quality, as surfel-based representations struggle to preserve high-frequency details. TextureGS [16] attempts to decouple appearance and geometry within the 2DGS framework, but the single representation still limits overall rendering performance. Recently, HybridGS [17] combines 3DGS with image-space 2D Gaussians to address transient objects, but its radiance field remains fully represented by 3D Gaussians. HorizonGS [18], designed for varying-altitude scenes, decodes 2D Gaussians for surface reconstruction and 3D Gaussians for view synthesis separately via an MLP in ScaffoldGS [22]. While effective in their target domains, these methods do not explore a unified hybrid radiance representation. As a result, the complementary strengths of 2DGS and 3DGS in geometry and appearance remain underutilized.

Effectively combining 3D and 2D Gaussians to jointly improve appearance and geometry is non-trivial, as simply mixing the two representations does not necessarily improve reconstruction quality [18]. To start, the geometric accuracy of 2D Gaussians relies on a ray–splat–intersection-based rasterizer designed to enforce multi-view consistency. Using the projection-based 3DGS rasterizer to render 2D Gaussians can lead to suboptimal geometry [15]. Moreover, Gaussian parameters change significantly during training. For instance, 3D Gaussians may flatten to approximate surfaces, while 2D Gaussians may expand volumetrically to capture thin structures or translucent effects. Fixing the Gaussian type throughout optimization can limit the model's expressiveness. Finally, relying solely on photometric loss is insufficient to balance geometry and appearance. Additional regularization is required to guide the optimization of hybrid representations. Most importantly, the regularization strategy should account for the distinct characteristics of 3D and 2D Gaussians.

In response to these challenges, we introduce **E**xchan**g**eable **G**aussian **S**platting (EGGS), an adaptive hybrid representation that unifies 2D and 3D Gaussian splatting in a single framework. EGGS provides a practical and efficient solution for high-quality novel view synthesis and 3D reconstruction. Our main contributions are as follows:

- To preserve the complementary strengths of 3D and 2D Gaussians, we develop *Hybrid Gaussian Rasterization*, a unified rendering framework that supports both projection-based and ray–splat–intersection-based rasterization. We implement this framework with CUDA for efficient optimization, and ensure compatibility with existing 3DGS and 2DGS pipelines.

- We propose *Adaptive Type Exchange*, which enables an exchangeable hybrid of 2D and 3D Gaussians. We use effective rank as an auxiliary criterion to determine whether each Gaussian should dynamically switch its type during training, resulting in a more flexible and content-adaptive representation.

- To better balance geometry and appearance, we introduce *Frequency-Decoupled Optimization*, a regularization strategy in the frequency domain. Using the Discrete Wavelet Transform (DWT), we extract low-frequency components to guide scene geometry and high-frequency components to refine appearance. We supervise 3D and 2D Gaussians asymmetrically to exploit their distinct characteristics, where high-frequency signals guide 3D Gaussians toward detailed appearance, while low-frequency signals supervise 2D Gaussians for geometric consistency.
- We conduct extensive experiments demonstrating that EGGS significantly improves the trade-off between appearance fidelity and geometric accuracy. It outperforms both 3DGS and 2DGS in appearance quality, while achieving geometric accuracy and multi-view consistency comparable to 2DGS. Moreover, EGGS serves as a versatile representation that performs well in challenging scenarios such as few-shot and out-of-distribution view synthesis.

## 2    Related Works

**Radiance Fields for Novel View Synthesis.** Neural Radiance Fields (NeRF) [4] have emerged as a fundamental approach for novel view synthesis [23], representing scenes as continuous volumetric functions optimized via differentiable rendering. While NeRF achieves high-fidelity reconstruction, it requires dense sampling and significant computational resources. Subsequent works have improved either quality [24, 25] or efficiency [5, 7, 26, 6], but the excessive training and rendering time remains a major bottleneck. To address this, recent efforts have explored more efficient alternatives, such as 3D Gaussian Splatting (3DGS) [11], which represents scenes using a set of 3D Gaussians that can be efficiently rasterized and optimized for real-time rendering. To further improve the performance and efficiency of 3DGS, several extensions have been proposed. ScaffoldGS [22] introduces a voxel-based representation where an MLP is used to decode 3D Gaussians within each voxel. 3DGS-MCMC [27] formulates Gaussian densification as a Markov Chain Monte Carlo sampling process, enabling a more efficient and adaptive distribution of Gaussians across the scene.

**Geometry-Appearance-Balanced Gaussian Splatting.** While 3DGS achieves high appearance fidelity and is efficient in both training and rendering, the anisotropic nature of 3D Gaussians often exhibits multi-view inconsistency, resulting in limited geometric accuracy. To address this, several works propose geometry regularization techniques. SUGAR [14] and GaussianPro [13] introduce normal-based regularization to encourage flatter Gaussians that better align with scene surfaces. Gaussian Surfels [15] and GOF [28] further enforce depth accuracy and normal consistency to enhance geometric reconstruction. Instead of relying solely on regularization, 2DGS [12] adopts a 2D surfel representation with a specialized ray–splat–intersection rasterizer, ensuring multi-view consistency and significantly improving geometric accuracy compared to 3DGS. It also incorporates additional depth and normal regularization. However, this comes at the cost of reduced appearance quality, as 2D surfels struggle to preserve high-frequency detail. TextureGS [16] attempts to decouple geometry and appearance modeling within the 2DGS framework, but its appearance fidelity remains limited due to the inherent drawbacks of the 2D representation.

As demonstrated in Figure 2 (left), 3D Gaussians achieve better appearance quality in PSNR, SSIM, and LPIPS. In contrast, 2D Gaussians offer superior view consistency and geometric fidelity, resulting in more robust PSNR under out-of-distribution (OOD) conditions, improved point cloud accuracy in Chamfer Distance (CD), and higher depth accuracy in F1 score. As shown in Figure 2 (right), most existing methods [12, 11, 14, 13, 29, 30, 22, 31, 32, 33, 34] rely on a single Gaussian representation to reconstruct radiance fields, which limits their flexibility and adaptability. Although HybridGS [17] incorporates both 3D Gaussians and image-space 2D Gaussians to better handle transient content, its radiance field remains solely represented by 3D Gaussians. A radiance field that jointly leverages both 2D and 3D Gaussians remains largely unexplored. It is still unclear how 2D and 3D Gaussians can be made exchangeable during training and how to fully exploit their complementary strengths in appearance and geometry. We provide a more detailed discussion in Appendix B.

## 3    Method

We provide an overview of the EGGS framework in Figure 3. To enable the joint training of 2D and 3D Gaussians within a unified framework, we first introduce *Hybrid Gaussian Rasterization* in Section 3.1, which supports both ray–splat–intersection-based rendering for 2D Gaussians and projection-based rendering for 3D Gaussians. Next, we present *Adaptive Type Exchange* in Section 3.2,

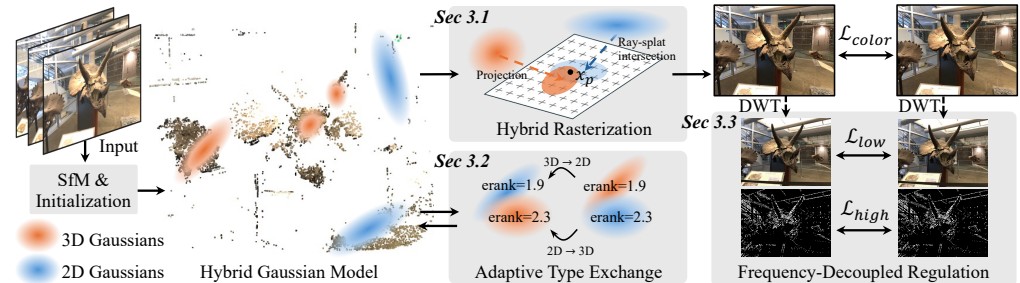

Figure 3: Overview of the EGGS framework. We initialize 2D and 3D Gaussians from sparse points obtained via structure-from-motion (SfM) [35, 36]. Their parameters are then jointly optimized using our CUDA-accelerated differentiable hybrid rasterization. To enhance the flexibility of the hybrid representation, Adaptive Type Exchange is introduced to allow each Gaussian to switch between 2D and 3D types during training. Finally, we apply Discrete Wavelet Transform (DWT) [37] and introduce Frequency-Decoupled Optimization to balance geometric accuracy and appearance fidelity.

which enables dynamic switching between 2D and 3D types during optimization. Finally, to optimize the hybrid model for balanced geometric consistency and appearance fidelity, we propose *Frequency-Decoupled Optimization* in Section 3.3, a supervision strategy that leverages the distinct frequency characteristics of 2D and 3D Gaussians.

## 3.1 Hybrid Gaussian Rasterization

Differentiable rasterization was introduced in 3DGS to enable gradient-based optimization of Gaussian parameters using a projection-based pipeline for real-time rendering. 2DGS later developed a ray–splat–intersection-based rasterizer tailored to 2D surfel representations, improving multi-view consistency and geometric accuracy. However, the architectural distinction between these two rasterization pipelines makes it non-trivial to render and optimize a hybrid model within a unified framework. While 2D Gaussians can be viewed as degenerate 3D Gaussians with zero scale along the $z$-axis, directly rendering them with the 3D rasterizer leads to geometric inaccuracies [12]. This is due to the affine projection approximation used in 3DGS, which introduces distortion at all points except the Gaussian center. We further analyze this issue in Section 4.2.

To leverage the complementary strengths of 3D and 2D Gaussians, it is necessary to render them within a unified framework. To this end, we propose *Hybrid Gaussian Rasterization*, which integrates both projection-based and ray–splat–intersection-based pipelines. In our rasterizer, each Gaussian primitive $\mathcal{G}$ is parameterized by a center $\boldsymbol{\mu} \in \mathbb{R}^3$, scale $\boldsymbol{s} \in \mathbb{R}^3$, rotation quaternion $\boldsymbol{r} \in \mathbb{R}^4$, opacity $\alpha \in \mathbb{R}$, and spherical harmonic (SH) color coefficients $\boldsymbol{f} \in \mathbb{R}^{3 \times (l+1)^2}$, where $l$ is the degree of view-dependent color. The view-dependent RGB color $\boldsymbol{c}$ is decoded from $\boldsymbol{f}$. The Gaussian shape is defined by the covariance matrix $\boldsymbol{\Sigma} = \boldsymbol{R}\boldsymbol{S}\boldsymbol{S}^T\boldsymbol{R}^T$, where $\boldsymbol{R} \in \mathbb{R}^{3 \times 3}$ is the rotation matrix derived from $\boldsymbol{r}$, and $\boldsymbol{S} = \text{diag}(s_x, s_y, s_z) \in \mathbb{R}^{3 \times 3}$ is the scaling matrix. We augment each Gaussian with a type specifier $t \in \{0, 1\}$ to indicate whether it is a 2D ($t = 0$) or 3D ($t = 1$) Gaussian. 2D Gaussians are initialized with $s_z = 0$, while the remaining parameters follow the initialization of 3DGS.

As shown in Figure 4, we rasterize Gaussians according to their types, where affine projection is used for 3D Gaussians and ray–splat–intersection is used for 2D Gaussians. The contribution of $\mathcal{G}_i^{3d}$ and $\mathcal{G}_i^{2d}$ is evaluated by computing the distance from the image-space pixel $x_p$ to $\mathcal{G}_i^{proj}$ or $\mathcal{G}_i^{2d}$ in the 2D image plane or tangent frame, respectively:

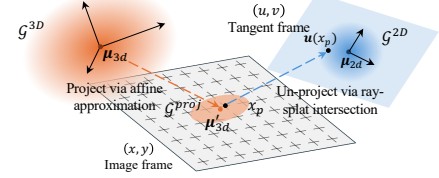

$$d_i = \begin{cases} (\boldsymbol{u}_i(x_p)^2 + \boldsymbol{v}_i(x_p)^2) & \text{if } t_i = 0 \\ (x_p - \boldsymbol{\mu}'_{3d,i})^T \boldsymbol{\Sigma}'^{-1}_i (x_p - \boldsymbol{\mu}'_{3d,i}) & \text{otherwise} \end{cases} \quad (1)$$

where $\boldsymbol{\mu}'_{3d,i}$ and $\boldsymbol{\Sigma}_i$ are the projected center and

Figure 4: Illustration of Hybrid Gaussian Rasterization. The contribution of 3D Gaussians and 2D Gaussians is computed via affine projection and ray-splat-intersection, respectively.

covariance of the 3D Gaussian computed via affine projection, and $u_i(x_p)$ and $v_i(x_p)$ denote the coordinates of the intersection between the ray through $x_p$ and the 2D Gaussian. The distance $d_i$ can be computed simultaneously for both 3D and 2D Gaussians. The final contribution of each Gaussian

is then computed uniformly as $\tilde{\alpha}_i = \alpha_i e^{-\frac{1}{2}d_i}$, where $\alpha_i$ is the opacity of the $i$-th Gaussian. With the above formulation, both 3D and 2D Gaussians can be rendered in a single $\alpha$-blending pass:

$$C(x_p) = \sum_{i \in N} \boldsymbol{c}_i \tilde{\alpha}_i \prod_{j=1}^{i-1}(1 - \tilde{\alpha}_j) \qquad (2)$$

where the final color at pixel $x_p$ is computed from color $\boldsymbol{c}_i$ and contribution $\tilde{\alpha}_i$ of each Gaussian primitive. To support efficient and parallel rendering, we implement our hybrid rasterizer in CUDA. More details on initialization and densification are provided in Appendix A, and those on projection-based and ray–splat–intersection-based rasterization procedures are deferred to Appendix C.

## 3.2 Adaptive Type Exchange

While the type specifier introduced in Section 3.1 enables unified rendering of 2D and 3D Gaussians, each Gaussian primitive is initialized with a fixed type. Such fixed type assignment can limit the expressiveness of the model, as Gaussians may naturally deviate from their initial type during optimization. For example, 3D Gaussians may become increasingly flat to better model surfaces, while 2D Gaussians may take on more volumetric properties to capture semi-transparent regions. To fully exploit the flexibility of the hybrid model, the type of each Gaussian should dynamically adapt to its evolving geometric characteristics. To this end, we propose *Adaptive Type Exchange*, which allows each Gaussian to switch between 2D and 3D types during training.

The key to *Adaptive Type Exchange* is detecting discrepancies between a Gaussian's assigned type and its effective geometric dimensionality. Therefore, we introduce the effective rank (erank) [38, 39] as an indicator of this dimensionality, allowing the model to determine when type switching is needed during training. Given a Gaussian $\mathcal{G}$ with scaling $\boldsymbol{s} = (s_x, s_y, s_z)$, we define its effective rank as:

$$\text{erank}(\mathcal{G}) = \exp\left(-\sum_{i=0}^{2} \frac{q_i}{\|\boldsymbol{q}\|_1} \log \frac{q_i}{\|\boldsymbol{q}\|_1}\right), \quad \text{where } \boldsymbol{q} = (s_x^2, s_y^2, s_z^2). \qquad (3)$$

As illustrated in Figure 5, erank provides a principled signal for deciding when to switch types. A perfectly isotropic 3D Gaussian has erank $= 3$, while a flattened Gaussian approaches erank $= 2$. If a Gaussian primitive $\mathcal{G}_i$ is assigned as 3D ($t_i = 1$) but its effective rank falls below a threshold $\theta_e$, we mark it for conversion to 2D by setting $t'_i = 0$. Similarly, we update 2D Gaussians to 3D ($t'_i = 1$) when their effective rank exceeds the threshold. Yet, merely flipping the type specifier can lead to unstable parameter transitions, as the $s_z$ scale is treated as least significant in 2D Gaussians. To ensure stable conversion, we reparameterize the covariance of 3D Gaussians during switching and adjust gradient flow to $s_z$ for 2D Gaussians.

**Reparameterization.** ($3D \rightarrow 2D$) All three scales of a 3D Gaussian are initially optimized, whereas 2D Gaussians ignore the $s_z$ scale during ray–splat–intersection-based rasterization. Accordingly, when converting a 3D Gaussian to 2D, only $s_x$ and $s_y$ are retained and $s_z$ is discarded. However, directly discarding $s_z$ can lead to instability during training when it is not the least significant scale. To prevent this, we reparameterize the 3D Gaussian before conversion so that $s_z$ corresponds to the smallest axis. The key to stable conversion is aligning $s_z$ with the least significant scale while preserving covariance $\boldsymbol{\Sigma} = \boldsymbol{R}\boldsymbol{S}\boldsymbol{S}^T\boldsymbol{R}^T$. We first construct the converted scaling matrix $\boldsymbol{S}^*$ using a permutation matrix $\boldsymbol{P}$ that moves the least significant scale to the $z$-axis:

$$\boldsymbol{S}^* = \boldsymbol{P}\boldsymbol{S}\boldsymbol{P}^T \qquad (4)$$

$\boldsymbol{P}$ is set to $\boldsymbol{P}_x$ or $\boldsymbol{P}_y$ if $s_x$ or $s_y$ is the least significant scale, respectively, where:

$$\boldsymbol{P}_x = \begin{bmatrix} 0 & 1 & 0 \\ 0 & 0 & 1 \\ 1 & 0 & 0 \end{bmatrix}, \quad \boldsymbol{P}_y = \begin{bmatrix} 0 & 0 & 1 \\ 1 & 0 & 0 \\ 0 & 1 & 0 \end{bmatrix} \qquad (5)$$

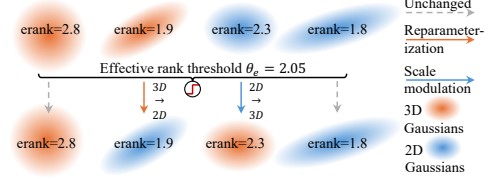

Figure 5: Illustration of Adaptive Type Exchange.

Then, to ensure the covariance $\boldsymbol{\Sigma}$ remain unchanged in $\boldsymbol{\Sigma}^* = \boldsymbol{R}^*\boldsymbol{S}^*\boldsymbol{S}^*\boldsymbol{R}^{*T}$, we set the rotation of 2D Gaussian as $\boldsymbol{R}^* = \boldsymbol{R}\boldsymbol{P}^T$. We note that $\boldsymbol{R}^*$ is converted to quaternions during optimization. To ensure a valid conversion, the rotation matrix $\boldsymbol{R}^*$ must be orthogonal with a positive determinant. $\boldsymbol{P}_x$ and $\boldsymbol{P}_y$ are designed to preserve these properties. Additional details are provided in Appendix D.

**Scale Modulation.** ($2D \rightarrow 3D$) When converting 2D Gaussians to 3D, all parameters are retained with the type specifier flipped. However, as mentioned above, the $s_z$ scale of 2D Gaussians is not

optimized during rasterization. However, to allow 2D Gaussians to develop volumetric capacity and transition to 3D when needed, gradient flow must also be introduced along the $z$-axis. To support this, we incorporate $s_z$ into the computation graph via a soft modulation based on opacity $\alpha$, aligning geometric expressiveness with visual transparency. The intuition behind this design is that 2D-to-3D transitions often occur in regions with semi-transparent or volumetric effects that flat primitives cannot represent well. This modulation enables $s_z$ to be optimized throughout training, while ensuring its updates remain stable and smoothly conditioned on opacity:

$$\alpha_i^* = \alpha_i e^{-\lambda_z \times s_z^*} \tag{6}$$

where $s_z^*$ denotes the activated $z$-axis scale, computed via a soft gating function:

$$s_z^* = \text{sigmoid}\left(\frac{s_z - \theta_z}{T_z}\right) s_z \tag{7}$$

The soft scale modulation allows a 2D Gaussian to remain effectively two-dimensional when $s_z$

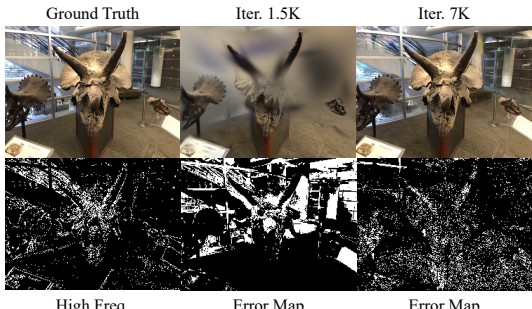

Ground Truth     Iter. 1.5K     Iter. 7K

High Freq.     Error Map     Error Map

Figure 6: Illustration of reconstruction error during training. In early iterations, the model focuses on overall scene geometry, while high-frequency local details are progressively refined in later stages.

is insignificant, in which case $s_z^*$ approaches zero. Conversely, as a 2D Gaussian evolves toward a more volumetric form, an increasing $s_z$ leads to reduced opacity, effectively enabling the representation of semi-transparent or volumetric effects. Additional details on the effective rank threshold, 3D Gaussian reparameterization and permutation, and 2D Gaussian scale modulation are provided in Appendix D.

## 3.3 Frequency-Decoupled Optimization

With our hybrid rasterization and adaptive type exchange mechanism, 2D and 3D Gaussians can be jointly optimized within a unified and flexible framework. However, relying solely on photometric loss is insufficient to effectively optimize the hybrid model for balanced geometry and appearance. 2D and 3D Gaussians exhibit distinct characteristics during optimization and specialize in different aspects of the scene. 2D Gaussians are better suited for enforcing geometric consistency, while 3D Gaussians excel at capturing high-frequency appearance details. To fully leverage these complementary strengths, we introduce *Frequency-Decoupled Optimization*, a supervision strategy that decouples low- and high-frequency components and assigns them asymmetrically to 2D and 3D Gaussians, respectively.

**Frequency Decoupling via Discrete Wavelet Transform.** As shown in Figure 6, scene information can be effectively separated in the frequency domain. High-frequency components typically correspond to fine details that are refined in later training stages (e.g., 7K iterations), while low-frequency components capture overall scene geometry and are optimized earlier. This frequency-based separation aligns well with the complementary roles of 3D Gaussians in modeling appearance and 2D Gaussians in capturing geometry. Motivated by this, we introduce *Frequency-Decoupled Optimization* to supervise the hybrid model in the frequency domain. We apply DWT [37] to decompose the ground truth image $\mathcal{I}$ into low- and high-frequency components: $\mathcal{I}_l, \mathcal{I}_h = \text{DWT}(\mathcal{I})$. The same transformation is applied to the rendered image $\hat{\mathcal{I}}$ to obtain $\hat{\mathcal{I}}_l$ and $\hat{\mathcal{I}}_h$.

The frequency loss is defined as $\mathcal{L}_i = \|\hat{\mathcal{I}}_i - \mathcal{I}_i\|_2^2$ for $i \in \{\text{low}, \text{high}\}$. We also include the standard appearance loss used in 3DGS: $\mathcal{L}_{\text{color}} = (1 - \lambda)\mathcal{L}_1 + \lambda\mathcal{L}_{\text{D-SSIM}}$. With access to frequency-specific losses, a naïve strategy is to combine all terms and apply them uniformly to all Gaussians:

$$\mathcal{L} = \mathcal{L}_{\text{color}} + \lambda_{\text{low}}\mathcal{L}_{\text{low}} + \lambda_{\text{high}}\mathcal{L}_{\text{high}}. \tag{8}$$

where $\mathcal{L}_{\text{low}}$ and $\mathcal{L}_{\text{high}}$ are applied equally to all 2D and 3D Gaussians. While supervision is decoupled in the frequency domain, this approach overlooks the distinctions of each representation.

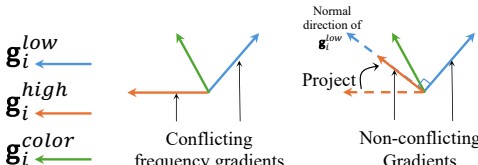

$\mathbf{g}_i^{low}$
$\mathbf{g}_i^{high}$
$\mathbf{g}_i^{color}$

Conflicting frequency gradients     Non-conflicting Gradients

Figure 7: Illustration of frequency gradient projection. We project gradients from high-frequency loss onto the normal vector of gradients from low-frequency for 2D Gaussians.

**Asymmetrical Gradient Update with Projected Conflicts.** We denote the gradients to $\mathcal{G}_i$ from $\mathcal{L}_{\text{color}}, \mathcal{L}_{\text{low}}$ and $\mathcal{L}_{\text{high}}$ as $\mathbf{g}_i^{color}, \mathbf{g}_i^{low}$ and $\mathbf{g}_i^{high}$, respectively. As illustrated in Figure 7, conflicting gradients can arise when losses from different frequency components are directly applied to update Gaussian parameters (i.e., Eq.(8)), diminishing the effectiveness of frequency-based regularization. Such conflicts stem from the distinct characteristics of 2D and 3D Gaussians. 2D Gaussians are more effective at capturing overall geometry and ensuring multi-view consistency, where low-frequency signals offer more relevant guidance, while high-frequency gradients may counteract this by encouraging appearance-driven updates. Conversely, 3D Gaussians specialize in modeling fine-scale appearance and benefit more from high-frequency supervision, whereas low-frequency signals contribute less to their performance.

To address this issue, we propose an asymmetrical update strategy that applies frequency supervision based on Gaussian type, as shown in Algorithm 1. For each Gaussian, we check for potential gradient conflicts by computing the inner product between $\mathbf{g}_i^{low}$ and $\mathbf{g}_i^{high}$, where a negative value indicates divergent update directions [40]. When such conflict is detected, we retain the frequency component most relevant to the Gaussian type and project the other. Specifically, for 2D Gaussians, we preserve supervision from low-frequency and remove the conflicting component of high-frequency by projecting $\mathbf{g}_i^{high}$ onto the normal vector of $\mathbf{g}_i^{low}$, as shown in Eq.(9). Similarly, for 3D Gaussians, we retain $\mathbf{g}_i^{high}$ and project $\mathbf{g}_i^{low}$ as indicated in Eq.(10). This asymmetrical supervision ensures each Gaussian is updated along its most informative direction while minimizing interference from less relevant frequency signals. More details on DWT and the gradient projection strategy are provided in Appendix E and Appendix F, respectively.

---

**Algorithm 1:** Frequency-Decoupled Optimization

**Require :** Gaussians $\{\mathcal{G}_i\}_{i=0}^{N-1}$, appearance loss $\mathcal{L}_{color}$, frequency loss $\mathcal{L}_{low}$ and $\mathcal{L}_{high}$.

$\mathbf{g}^{color}, \mathbf{g}^{low}, \mathbf{g}^{high} \leftarrow \nabla_\mathcal{G} \mathcal{L}_{color}, \nabla_\mathcal{G} \mathcal{L}_{low}, \nabla_\mathcal{G} \mathcal{L}_{high}$;
// Process per Gaussian gradient conflict
**for** $i \leftarrow 0$ **to** $N-1$ **do**
  // There are conflict Gradients in different frequencies
  **if** $\mathbf{g}_i^{low} \cdot \mathbf{g}_i^{high} < 0$ **then**
    **if** $t_i == 0$ **then**
$$\mathbf{g}_i^{high} \leftarrow \mathbf{g}_i^{high} - \frac{\mathbf{g}^{high} \cdot \mathbf{g}^{low}}{||\mathbf{g}^{low}||^2} \mathbf{g}^{low}; \qquad (9)$$
      // Type is 2D, project $\mathbf{g}_i^{high}$ onto normal of $\mathbf{g}_i^{low}$
    **else**
$$\mathbf{g}_i^{low} \leftarrow \mathbf{g}_i^{low} - \frac{\mathbf{g}^{high} \cdot \mathbf{g}^{low}}{||\mathbf{g}^{high}||^2} \mathbf{g}^{high}; \qquad (10)$$
      // Type is 3D, project $\mathbf{g}_i^{low}$ onto normal of $\mathbf{g}_i^{high}$

  $\Delta \mathcal{G}_i = \mathbf{g}_i^{color} + \mathbf{g}_i^{low} + \mathbf{g}_i^{high}$
**return** Update $\Delta \mathcal{G}$

---

## 4 Experiments

**Datasets and Metrics.** We evaluate EGGS on several widely used benchmarks. For appearance evaluation, we use Mip-NeRF360 [25], LLFF [41], Tanks&Temples [42], and DTU [43]. For geometry evaluation, we use DTU, which provides ground-truth point clouds, and Tanks&Temples, which offers ground-truth depth maps. Additional dataset details are provided in Appendix A. Following prior work [25, 11, 13, 12], we report PSNR, SSIM [44], and LPIPS [45] to evaluate the appearance quality of synthesized novel views. For geometry, we follow [12, 39] and report Chamfer Distance [46] on DTU to assess reconstruction accuracy.

**Baselines.** To demonstrate the effectiveness of EGGS, we compare against several single-representation methods that use either 3D or 2D Gaussians. For 3D Gaussian-based methods, we include vanilla 3DGS [11], GaussianPro [13] and GOF [28], which incorporate geometric regularization, and FreGS [47], which introduces frequency-based supervision. For 2D Gaussian-based methods, we consider vanilla 2DGS [12] and TextureGS [16], which improves the appearance fidelity of 2D Gaussians. Additional discussion of related methods is provided in Appendix B.

**Implementation.** We implement our hybrid rasterizer based on the CUDA rasterization code of 3DGS [11]. We used the Haar filter for the DWT [37, 48]. For Frequency-Decoupled Optimization, we set the weight for the frequency components as $\lambda_{low} = 0.2$ and $\lambda_{low} = 0.4$. For Adaptive Type Switch, we set the erank threshold as $2.05$. We offer more details about our training pipeline and parameter setting in Appendix A. All experiments are conducted on an A5000 GPU.

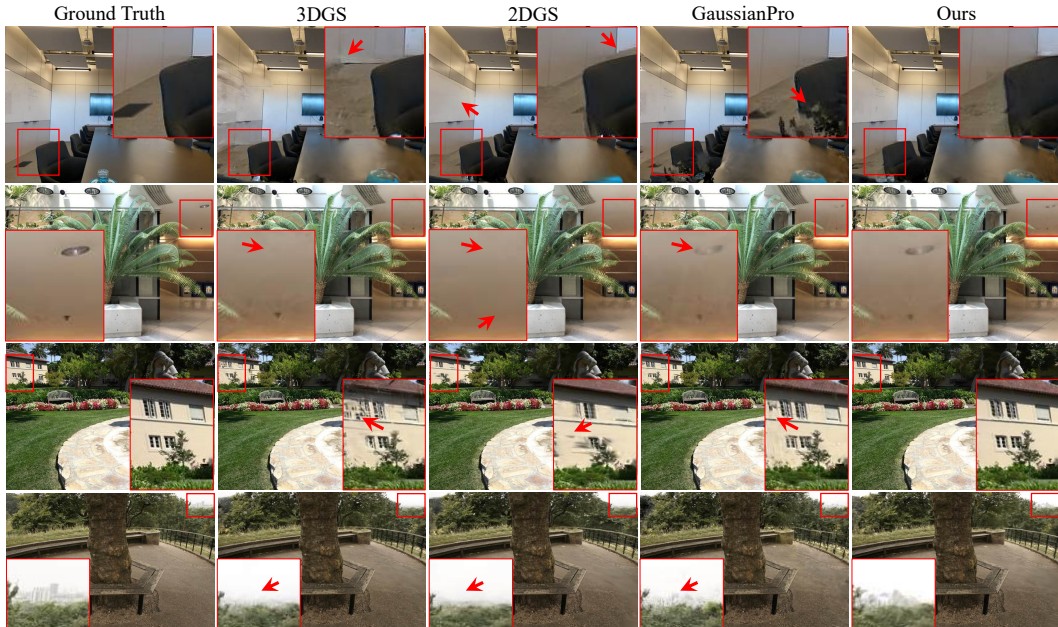

| Ground Truth | 3DGS | 2DGS | GaussianPro | Ours |

Figure 8: Qualitative comparison on LLFF, Tanks&Temples, and Mip-NeRF360. 3DGS suffers from inaccurate scene geometry. While 2DGS improves geometric fidelity, it overlooks texture and local details. EGGS recovers more accurate geometry while preserving high-frequency details. Additional visual results and videos are available in the supplementary material and project website.

Table 1: Quantitative comparison on Mip-NeRF360, LLFF and Tanks&Temples datasets. The best, second-best, and third-best entries are marked in red , orange , and yellow , respectively.

| Method | Mip-NeRF360 | | | LLFF | | | Tanks&Temples | | |
|---|---|---|---|---|---|---|---|---|---|
| | PSNR ↑ | SSIM ↑ | LPIPS ↓ | PSNR ↑ | SSIM ↑ | LPIPS ↓ | PSNR ↑ | SSIM ↑ | LPIPS ↓ |
| 2DGS $_{\text{SIGGRAPH'24}}$ [12] | 26.81 | 0.796 | 0.297 | 24.93 | 0.815 | 0.147 | 22.96 | 0.802 | 0.195 |
| TextureGS $_{\text{ECCV'24}}$ | 27.14 | 0.803 | 0.285 | 25.58 | 0.837 | 0.117 | 22.43 | 0.811 | 0.189 |
| 3DGS $_{\text{SIGGRAPH'23}}$ [11] | 27.43 | 0.814 | 0.257 | 26.12 | 0.865 | 0.099 | 23.85 | 0.833 | 0.168 |
| GOF $_{\text{TOG'24}}$ [28] | 27.42 | 0.826 | 0.234 | 25.57 | 0.854 | 0.121 | 22.41 | 0.831 | 0.172 |
| GaussianPro $_{\text{ICML'24}}$ [13] | 27.92 | 0.825 | 0.208 | 26.53 | 0.867 | 0.105 | 23.92 | 0.855 | 0.162 |
| FreGS $_{\text{CVPR'24}}$ [47] | 27.85 | 0.826 | 0.209 | 26.11 | 0.860 | 0.102 | 23.96 | 0.849 | 0.178 |
| Ours | 27.96 | 0.851 | 0.192 | 27.34 | 0.895 | 0.083 | 24.41 | 0.923 | 0.153 |

## 4.1 Results

**Appearance.** Figure 8 and Table 1 show qualitative and quantitative comparisons on Mip-NeRF360, LLFF, and Tanks&Temples. 3D Gaussian-based methods generally achieve better PSNR, SSIM, and LPIPS but often produce blurred geometry due to anisotropic Gaussians. GaussianPro and FreGS improve reconstruction via geometric or frequency regularization but still lack geometric accuracy. 2DGS produces cleaner edges and better geometry, yet oversmooths details and underperforms in appearance. TextureGS enhances 2D appearance but remains inferior to 3D-based methods. In contrast, EGGS outperforms all baselines by combining the strengths of 2D and 3D Gaussians. It recovers more accurate geometry while preserving high-frequency visual details.

**Geometry.** We evaluate geometry reconstruction quality on Tanks&Temples and DTU. As shown in Figure 9, both 2DGS and EGGS produce more accurate depth maps than 3DGS, with sharper surfaces and clearer edges. However, 2DGS sacrifices appearance fidelity due to the lack of high-frequency detail. In contrast, EGGS improves geometry over 3DGS while also preserving appearance quality. Table 2 reports Chamfer Distance on the DTU dataset, where EGGS outperforms 3DGS and SUGAR. Note that SUGAR, 2DGS, and GOF prioritize surface reconstruction and mesh extraction, often at the cost of appearance. Although 2DGS is slightly more accurate geometrically, EGGS achieves a better trade-off, offering stronger appearance quality alongside competitive geometry.

**Efficiency.** Table 3 compares the model size and training time of EGGS with 3DGS, 2DGS, and GaussianPro on LLFF and Tanks&Temples. While 2DGS uses the fewest Gaussians, its training time exceeds that of 3DGS. GaussianPro enhances appearance quality over 3DGS but incurs significantly higher training cost. In contrast, EGGS strikes a favorable balance, requiring fewer Gaussians than both 3DGS and GaussianPro, while achieving the shortest training time among all methods.

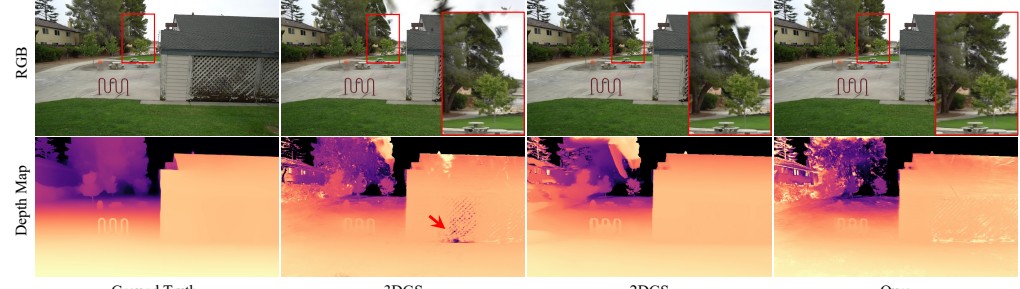

Figure 9: Qualitative comparison on Tanks&Temples. EGGS achieves better overall reconstruction quality, producing more accurate depth maps than 3DGS and recovering high-frequency details better than 2DGS.

Table 3: Comparison on efficiency. # Gaussians is the average number of Gaussians.

| Dataset | Method | PSNR | #Gaussians | Training |
|---|---|---|---|---|
| LLFF | 3DGS | 26.12 | 919K | 10min |
| | 2DGS | 24.93 | 343K | 11min |
| | GaussianPro | 26.53 | 933K | 20min |
| | EGGS | 27.34 | 581K | 9min |
| Tanks& Temples | 3DGS | 23.85 | 1502K | 13min |
| | 2DGS | 22.96 | 416K | 15min |
| | GaussainPro | 23.92 | 1381K | 35min |
| | EGGS | 24.41 | 754K | 11min |

Table 4: Ablation study. Repr. stands for the Gaussian type, Hyb. for hybrid rasterization, Ex. for type exchange, and Freq. for frequency regularization.

| ID | Repr. | Hyb. | Ex. | Freq. | PSNR↑ | SSIM ↑ | LPIPS ↓ |
|---|---|---|---|---|---|---|---|
| i | 3D | ✗ | ✗ | ✗ | 26.12 | 0.865 | 0.099 |
| ii | 3D+2D | ✗ | ✗ | ✗ | 26.01 | 0.859 | 0.105 |
| iii | 3D+2D | ✓ | ✗ | ✗ | 26.23 | 0.867 | 0.097 |
| iv | 3D+2D | ✓ | ✓ | ✗ | 26.58 | 0.874 | 0.093 |
| v | 3D | ✗ | ✗ | ✓ | 26.19 | 0.867 | 0.101 |
| vi | 3D+2D | ✓ | ✗ | ✓ | 26.41 | 0.871 | 0.096 |
| vii | 3D+2D | ✓ | ✓ | ✓ | 27.34 | 0.895 | 0.083 |

## 4.2 Ablation and Generalization Analysis

**Ablation Study.** We evaluate the effectiveness of each component in EGGS in Table 4. Row *i* is the vanilla 3DGS baseline. In row *ii*, we adopt a hybrid 2D/3D representation but rasterize all Gaussians using the 3DGS rasterizer, which leads to performance degradation. Row *iii* incorporates our hybrid rasterizer, which renders Gaussians according to their type. However, this setting still lacks flexibility and regularization. Row *iv* incorporates adaptive type exchange to enhance the flexibility. Rows *v–vii* study frequency-based supervision, which provides only limited gains for non-hybrid 3DGS (row *v*) but is more effective in the hybrid setting. The full model in row *vii* achieves the best performance, indicating that decoupled frequencies more effectively exploit the strengths of the exchangeable hybrid representation. We provide more ablation and analysis in Appendix F.

**Generalization Analysis.** We evaluate the robustness of EGGS in challenging scenarios, including few-shot and out-of-distribution (OOD) settings. Following prior work, we use LLFF [41] for few-shot evaluation [49, 50] and OOD-NVS [51] for OOD evaluation [52]. More details are provided in Appendix A. As shown in Table 11 and Figure 10, EGGS achieves robust performance in both settings, benefiting from its balanced multi-view consistency and appearance fidelity. This indicates that the hybrid representation generalizes better than single-type baselines.

We also emphasize that EGGS serves as a general underlying representation and is compatible with various optimization strategies developed for specialized settings [49, 50, 53, 54, 55, 51]. We discuss these orthogonal techniques in Appendix B, and provide further remarks on limitations and broader impacts in Appendix H.

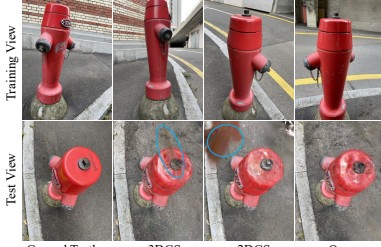

Figure 10: Comparison in the OOD setting.

Table 5: Generalization performance.

| Setting | Method | PSNR ↑ | SSIM ↑ | LPIPS ↓ |
|---|---|---|---|---|
| Few-shot | 3DGS | 19.52 | 0.719 | 0.279 |
| | 2DGS | 18.50 | 0.661 | 0.321 |
| | EGGS | 20.13 | 0.735 | 0.258 |
| OOD | 3DGS | 20.21 | 0.763 | 0.242 |
| | 2DGS | 23.52 | 0.863 | 0.188 |
| | EGGS | 24.17 | 0.907 | 0.151 |

Table 2: Quantitative Geometry Comparison on DTU. Chamfer Distance (CD) is reported per scene. mCD and PSNR denote the mean Chamfer Distance and mean PSNR across all scenes, respectively.

| Method | 24 | 37 | 40 | 55 | 63 | 65 | 69 | 83 | 97 | 105 | 106 | 110 | 114 | 118 | 122 | mCD | PSNR |
|---|---|---|---|---|---|---|---|---|---|---|---|---|---|---|---|---|---|
| 3DGS | 2.14 | 1.53 | 2.08 | 1.68 | 3.49 | 2.21 | 1.43 | 2.07 | 2.22 | 1.75 | 1.79 | 2.55 | 1.53 | 1.52 | 1.50 | 1.96 | 32.82 |
| SUGAR | 1.47 | 1.33 | 1.13 | 0.61 | 2.25 | 1.71 | 1.15 | 1.63 | 1.62 | 1.07 | 0.79 | 2.45 | 0.98 | 0.88 | 0.79 | 1.33 | 31.59 |
| 2DGS | 0.48 | 0.91 | 0.39 | 0.39 | 1.01 | 0.83 | 0.81 | 1.36 | 1.27 | 0.76 | 0.70 | 1.40 | 0.40 | 0.76 | 0.52 | 0.80 | 32.43 |
| GOF | 0.50 | 0.82 | 0.37 | 0.37 | 1.12 | 0.78 | 0.73 | 1.18 | 1.29 | 0.71 | 0.77 | 0.90 | 0.44 | 0.69 | 0.49 | 0.74 | 32.58 |
| Ours | 0.65 | 0.77 | 0.58 | 0.53 | 1.08 | 1.01 | 0.96 | 1.31 | 1.45 | 0.72 | 0.88 | 1.53 | 0.67 | 0.83 | 0.66 | 0.91 | 33.65 |

# 5  Conclusion

This paper presents EGGS, a hybrid Gaussian Splatting framework that combines the appearance fidelity of 3D Gaussians with the geometric accuracy of 2D Gaussians. The design integrates Hybrid Gaussian Rasterization for unified rendering, Adaptive Type Exchange for flexible representation, and Frequency-Decoupled Optimization to balance geometry and appearance. EGGS outperforms both 2D- and 3D-only baselines across multiple benchmarks. Future work includes extending the hybrid representation to more diverse and challenging scenarios.

**Acknowledgments**: This work was supported by Intelligence Advanced Research Projects Activity (IARPA) via Department of Interior/Interior Business Center (DOI/IBC) contract number 140D0423C0074. The U.S. Government is authorized to reproduce and distribute reprints for Governmental purposes, notwithstanding any copyright annotation thereon. Disclaimer: The views and conclusions contained herein are those of the authors and should not be interpreted as necessarily representing the official policies or endorsements, either expressed or implied, of IARPA, DOI/IBC, or the U.S. Government.

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

# Appendix

Table of contents:

## A   Implementation Details

**Training Pipeline and Parameter Setting.** Our training setup closely follows 3DGS [11]. We assume camera poses are provided or can be estimated using structure-from-motion (SfM) [35]. Initial sparse point clouds are generated via COLMAP [35, 36]. All methods, including EGGS and baselines, are trained for 30K iterations. Learning rates for Gaussian parameters follow the default configurations from 3DGS and 2DGS. We adopt the densification strategy from 3DGS, which refines Gaussian distributions by pruning or duplicating them in under- or over-reconstructed regions. Densification begins at iteration 500, ends at iteration 15K, and is performed every 100 iterations.

In EGGS, we employ Hybrid Gaussian Rasterization, where each Gaussian is rendered according to its assigned type. Gaussian types are randomly initialized. To enable flexible representation, we introduce Adaptive Type Exchange, which is performed every 500 iterations from step 500 to 30K. During type switching, we set the effective rank threshold $\theta_e$ to 2.05. For scale modulation, which allows 2D Gaussians to evolve into 3D Gaussians as the $s_z$ scale becomes more significant, we use $\theta_z = 1.05$ and temperature $T = 0.001$ in Eq. 7. Additionally, we incorporate frequency-based supervision using the Discrete Wavelet Transform (DWT). More details on Frequency-Decoupled Optimization are provided in Appendix F.

**Datasets and Evaluation Protocols.** We evaluate the performance of EGGS and baselines on several widely used datasets, including Mip-NeRF360 [25], LLFF [41], Tanks&Temples [42], DTU [43], and OOD-NVS [51]. We follow the standard train/test splits used in prior work [25], with additional dataset statistics provided in Table 6. To ensure fair comparison with baselines [11, 12, 28], we downsample input images using the same factors as in [52]. All datasets provide RGB images for evaluating appearance quality. DTU additionally offers ground-truth point clouds for computing Chamfer Distance, while Tanks&Temples provides ground-truth depth maps for depth accuracy evaluation. Beyond standard dense-view evaluation, we also perform evaluation under few-shot settings using 3 views from LLFF, and out-of-distribution (OOD) setting using OOD-NVS.

## B   Related Works

Following 3DGS [11], a line of work has been proposed to enhance the geometry accuracy and reconstruction quality. As discussed in the related work section, most existing methods are based on

Table 6: Details on the datasets used for evaluation of appearance and geometry.

| Dataset | Ground Truth | Evaluation Metric | Evaluation Protocol | Factor |
|---|---|---|---|---|
| Mip-NeRF 360 (outdoor) [25] | RGB image | Appearance | Standard | 4 |
| Mip-NeRF 360 (indoor) [25] | RGB image | Appearance | Standard | 2 |
| LLFF [41] | RGB image | Appearance | Standard and Few-shot | 8 |
| Tanks&Temples [42] | RGB image; Depth | Appearance and Geometry (F1) | Standard | 2 |
| DTU [43] | RGB image; Point Cloud | Geometry (chamfer distance) | Standard | 2 |
| OOD-NVS [51] | RGB image | Appearance | OOD | 1 |

a single representation [14, 13, 12, 15, 28, 16, 47]. Here, we further expand on several recent efforts that aim to exploit the advantages of 3D and 2D Gaussians, as summarized in Table 7. We consider a method to be general if the training pipeline follows the standard 3DGS.

HybridGS [17] proposes to combine 3D Gaussians and image-space 2D Gaussians to remove transient objects during reconstruction. However, the 2D Gaussians in HybridGS are defined in the image frame, lacking the multi-view consistency provided by 2DGS [12]. Additionally, HybridGS employs a three-stage training pipeline specifically designed for transient object removal, making it less general than pipelines based on 3DGS. Moreover, since the training code of HybridGS is not publicly available, a direct comparison with EGGS is not feasible.

HorizonGS [18], on the other hand, is built upon ScaffoldGS, where voxel-based MLPs are used to decode Gaussian primitives. HorizonGS generates 3D Gaussians from MLPs for novel view synthesis and 2D Gaussians for surface reconstruction. Thus, it still follows a single-representation scheme. Furthermore, HorizonGS is specifically designed for aerial-ground scenarios and introduces a two-stage training pipeline to address conflicts from varying altitudes. Similar to HybridGS, this design is task-specific and less generalizable.

In contrast to HybridGS and HorizonGS, we note that several recent optimization techniques are more general and adhere to the 3DGS training pipeline, such as ScaffoldGS [22] and 3DGS-MCMC [27]. ScaffoldGS improves efficiency by introducing voxel-based MLPs, while 3DGS-MCMC enhances the densification process by reformulating 3D Gaussians as Markov Chain Monte Carlo (MCMC) samples. These methods are orthogonal to EGGS. Since EGGS serves as a general underlying representation, we believe such orthogonal optimizations can be incorporated to further enhance our framework. Exploring the potential of integrating these techniques with our exchangeable hybrid representation is a promising direction for future work.

In this work, we assume that camera poses are either available or can be estimated using structure-from-motion (SfM) [35], and that initial sparse point clouds can be generated using COLMAP [35, 36]. However, in practice, recovering geometric information such as camera poses and point clouds remains challenging. Recently, 3D Geometric Foundation Models (GFMs) have emerged as a promising approach to improve the generalizability of 3D reconstruction [56]. Feed-forward models such as DUSt3R [57], MASt3R [58], and VGGT [59] can predict robust geometric attributes in a single forward pass, even when the input multi-view images exhibit minimal or no overlap. Incorporating GFMs can potentially enhance the reconstruction quality of our method in open-world scenarios.

Table 7: Comparison of the setting with related works.

| Method | Gaussian Type | Setting | Training Pineline |
|---|---|---|---|
| 3DGS [11] | 3D | General | Single-stage |
| 2DGS [12] | 2D | General | Single-stage |
| HybridGS [17] | 3D+2D | Transient | Three-stage |
| HorizonGS [18] | 3D/2D | Varying-altitude | Two-stage |
| ScaffoldGS [22] | 3D | General | Single-stage |
| 3DGS-MCMC [27] | 3D | General | Single-stage |
| EGGS | 3D+2D | General | Single-stage |

## C Differentiable Rasterization for 2D/3D Gaussians Splatting

In this section, we detail the differentiable rasterization procedures used in 3DGS [11] and 2DGS [12], focusing on how the intermediate coordinates in Eq.(1) are computed. In both methods, images are rendered by computing the color at each screen-space pixel $x_p$ from a set of $N$ Gaussians $\{\mathcal{G}_i\}_{i=0}^{N-1}$. The final pixel color is obtained by rasterizing the Gaussians onto the image plane, but the rasterization procedures differ significantly between 3DGS and 2DGS.

3DGS employs an affine approximation to the projective transformation for rasterization. For a 3D Gaussian $\mathcal{G}_i^{3d}$ with type $t_i = 1$, it is first projected onto the image plane via affine projection [11]. The resulting projected Gaussian is denoted as $\mathcal{G}_i^{\text{proj}}$, with 2D center $\boldsymbol{\mu}_{3d,i}'$ and covariance $\boldsymbol{\Sigma}_i'$ in the image plane. The projected covariance is computed as:

$$\boldsymbol{\Sigma}_i' = \boldsymbol{JW}\boldsymbol{\Sigma}_i\boldsymbol{W}^T\boldsymbol{J}^T, \tag{11}$$

where $\boldsymbol{J}$ is the Jacobian matrix of the affine projection, and $\boldsymbol{W}$ accounts for the world-to-camera transformation [11]. Once $\boldsymbol{\Sigma}_i'$ is obtained, the projected center $\boldsymbol{\mu}_{3d,i}'$ can be computed accordingly.

On the other hand, 2DGS applies ray–splat–intersection to rasterize 2D Gaussians. For a 2D Gaussian $\mathcal{G}_i^{2d}$ with $t_i = 0$, it is not directly projected onto the image plane. Instead, to preserve the geometric accuracy of $\mathcal{G}_i^{2d}$, the pixel $x_p$ is unprojected into the local tangent frame defined by the Gaussian [12]. This is done by computing the intersection between the ray passing through $x_p$ and the tangent plane of $\mathcal{G}_i^{2d}$, resulting in the local coordinates:

$$u(x_p) = \frac{h_u^2 h_v^4 - h_u^4 h_v^2}{h_u^1 h_v^2 - h_u^2 h_v^1}, \quad v(x_p) = \frac{h_u^4 h_v^1 - h_u^1 h_v^4}{h_u^1 h_v^2 - h_u^2 h_v^1}, \tag{12}$$

where $h_u$ and $h_v$ are derived from the homogeneous plane equations associated with the pixel $x_p = (x, y)$ as:

$$h_u = (\boldsymbol{WH})^T h_x, \quad h_v = (\boldsymbol{WH})^T h_y. \tag{13}$$

More details about the homogeneous transformation matrices $\boldsymbol{H}$ and $\boldsymbol{W}$ can be found in 2DGS [12]. By solving Eq. (12), we obtain the 2D position of $x_p$ in the tangent frame, denoted as $\mathbf{u}_i(x_p) = (\boldsymbol{u}_i(x_p), \boldsymbol{v}_i(x_p))$. Note that the center $\boldsymbol{\mu}_{2d,i}$ of $\mathcal{G}_i^{2d}$ is defined as the origin of this tangent frame.

With the projected center $\boldsymbol{\mu}_{3d,i}'$ and covariance $\boldsymbol{\Sigma}_i$ for 3D Gaussians, and the intersection coordinates $\boldsymbol{u}_i(x_p)$ and $\boldsymbol{v}_i(x_p)$ for 2D Gaussians, we can perform hybrid rasterization as described in Section 3.1.

## D Effective Rank and Type Switch

In this section, we provide additional details about Adaptive Type Exchange. We focus on the effective rank threshold and the design choices behind 3D Gaussian reparameterization and 2D Gaussian scale modulation. We also present statistics on the distribution of Gaussian types over training iterations.

**Effective Rank Threshold.** As described in Section 3.2, we use the effective rank (erank) to assess the mismatch between a Gaussian's assigned type and its actual geometric dimensionality. We set the erank threshold to 2.05 in our experiments and study its effect in Table 8.

Table 8: Ablation on different erank thresholds. We evaluate the appearance of EGGS on the LLFF dataset with different erank thresholds.

| erank threshold | PSNR | SSIM | LPIPS |
|---|---|---|---|
| 1.9 | 26.15 | 0.871 | 0.103 |
| 1.95 | 26.23 | 0.868 | 0.101 |
| 2 | 26.49 | 0.874 | 0.097 |
| 2.05 | 27.24 | 0.885 | 0.086 |
| 2.1 | 26.37 | 0.844 | 0.113 |
| 2.15 | 25.78 | 0.831 | 0.126 |

When a 3D Gaussian becomes increasingly flat and its erank drops below the threshold, it is converted to a 2D Gaussian. Conversely, if a 2D Gaussian's erank exceeds the threshold, it is switched to 3D. A higher threshold causes more 3D Gaussians to be converted to 2D, as more will fall below the threshold. This can degrade performance when the threshold is set too high. In contrast, a lower threshold (e.g., 1.9) results in fewer 3D Gaussians being converted to 2D, causing the model to behave more like the 3DGS baseline and limiting the benefits of hybrid representation.

While the erank metric has been previously introduced [38, 39], our contribution lies in its integration into a dynamic type exchange mechanism for hybrid Gaussian representation. We acknowledge that

erank is a heuristic measure of effective dimensionality and may not behave monotonically in all scenarios. When a Gaussian has scales $(1, 1, s_z)$ with the first two scales fixed, the erank increases with $s_z$ initially but may drop as $s_z$ becomes dominant (e.g., erank returns to 2 when $s_z \approx 2.6$). In such cases, a volumetric Gaussian could technically fall below the threshold $\theta_e$ and be converted to 2D. However, we note that such configurations are rare in practice. All three scales are updated jointly during training, and our method includes reparameterization and soft modulation to ensure that type transitions remain stable and consistent with the evolving shape. Although the erank threshold does not come with theoretical guarantees for all edge cases, it demonstrates empirical effectiveness across diverse datasets.

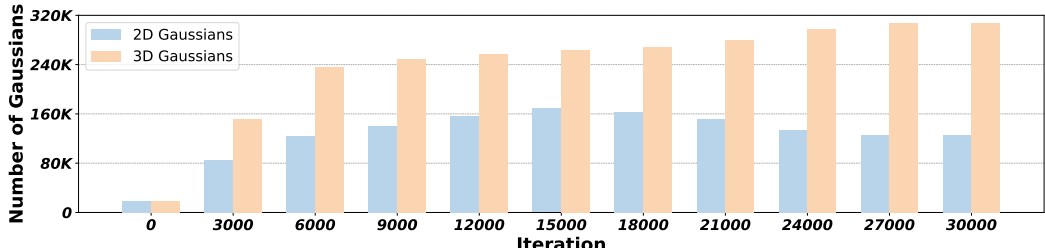

Figure 11: The number 3D and 2D Gaussians in different iterations during training.

**Permutation-Based Reparameterization.** The key to reparameterization is preserving the covariance during type switching. We achieve this using permutation matrices $\boldsymbol{P}_x$ and $\boldsymbol{P}_y$, designed to be orthogonal with a positive determinant. Specifically, we define:

$$\boldsymbol{P}_x = \begin{bmatrix} 0 & 1 & 0 \\ 0 & 0 & 1 \\ 1 & 0 & 0 \end{bmatrix}, \quad \boldsymbol{P}_y = \begin{bmatrix} 0 & 0 & 1 \\ 1 & 0 & 0 \\ 0 & 1 & 0 \end{bmatrix} \tag{14}$$

where $\boldsymbol{P}\boldsymbol{P}^T = \boldsymbol{I}$ and $\boldsymbol{I}$ is the identity matrix, and $\det(\boldsymbol{P}) = 1$. During conversion, the new $\boldsymbol{S}^*$ scaling and rotation $\boldsymbol{R}^*$ are computed using these permutation matrices to ensure that the transformed covariance $\boldsymbol{\Sigma} = \boldsymbol{R}\boldsymbol{S}\boldsymbol{S}\boldsymbol{R}^T$ remains unchanged:

$$\boldsymbol{S}^* = \boldsymbol{P}\boldsymbol{S}\boldsymbol{P}^T \text{ and } \boldsymbol{R}^* = \boldsymbol{R}\boldsymbol{P}^T \tag{15}$$

It is easy to see the converted covariance is unchanged:

$$\begin{aligned} \boldsymbol{\Sigma}^* &= \boldsymbol{R}^* \boldsymbol{S}^* \boldsymbol{S}^{*T} \boldsymbol{R}^{*T} \\ &= \boldsymbol{R}\boldsymbol{P}^T \cdot \boldsymbol{P}\boldsymbol{S}\boldsymbol{P}^T \cdot \boldsymbol{P}\boldsymbol{S}^T\boldsymbol{P}^T \cdot \boldsymbol{P}\boldsymbol{R}^T \\ &= \boldsymbol{R} \cdot (\boldsymbol{P}^T\boldsymbol{P}) \cdot \boldsymbol{S} \cdot (\boldsymbol{P}^T\boldsymbol{P}) \cdot \boldsymbol{S}^T \cdot (\boldsymbol{P}^T\boldsymbol{P}) \cdot \boldsymbol{R}^T \\ &= \boldsymbol{R}\boldsymbol{S}\boldsymbol{S}^T\boldsymbol{R}^T \\ &= \boldsymbol{\Sigma} \end{aligned} \tag{16}$$

In addition to preserving the covariance, the permutation must ensure that the converted rotation matrix $\boldsymbol{R}^*$ has a positive determinant. This is important because $\boldsymbol{R}^*$ is converted into a quaternion during optimization, and most 3D graphics frameworks assume right-handed coordinate systems [46]. A negative determinant implies a reflection, which cannot be represented by a unit quaternion. Given that the original rotation matrix $\boldsymbol{R}$ satisfies $\det(\boldsymbol{R}) > 0$, and the permutation matrices $\boldsymbol{P}_x$ and $\boldsymbol{P}_y$ are orthogonal with $\det(\boldsymbol{P}) = 1$, the determinant of the converted rotation $\boldsymbol{R}^* = \boldsymbol{R}\boldsymbol{P}^T$ is given by: $\det(\boldsymbol{R}^*) = \det(\boldsymbol{R}) \cdot \det(\boldsymbol{P}^T) = \det(\boldsymbol{R}) \cdot \det(\boldsymbol{P}) = \det(\boldsymbol{R}) > 0$. Therefore, $\boldsymbol{P}_x$ and $\boldsymbol{P}_y$ preserve both the covariance and the positive determinant of the rotation matrix, ensuring compatibility with quaternion-based optimization.

**Gaussian-Type Distribution.** Figure 11 shows the distribution of 2D and 3D Gaussians throughout training. As described in Appendix A, we randomly initialize Gaussian types, resulting in roughly equal numbers of 2D and 3D Gaussians at the start. During optimization, both types increase due to densification, with a more significant rise in 3D Gaussians. This trend can be attributed to the SfM-initialized points already containing geometric structure, allowing 2D Gaussians to capture coarse geometry in early iterations. Notably, the number of 2D Gaussians gradually decreases in

the later stages of training, especially after densification ends at 15K iterations. This suggests that many 2D Gaussians are being converted to 3D Gaussians, likely to better recover under-reconstructed regions and capture finer scene details.

**Gaussian-Type Initialization.** A key feature of EGGS is its exchangeable representation, which allows each Gaussian primitive to change its type as needed, regardless of its initial type. To verify this capability, we conduct a simple experiment by investigating three initialization scenarios: we initialize all Gaussians as 2D, all as 3D, or with random type assignments, and observe the distribution of Gaussian types throughout training. Below, we show the percentage of 3D Gaussians at different iterations. As shown in Table 9, even when the model is initialized entirely with 2D Gaussians, part of the Gaussians are converted to 3D Gaussians during training, leading to a hybrid model in the final stage. This demonstrates that 2D Gaussians can indeed transition to 3D types during training, despite potentially incorrect initialization, and vice versa.

# E   Discrete Wavalet Transformation

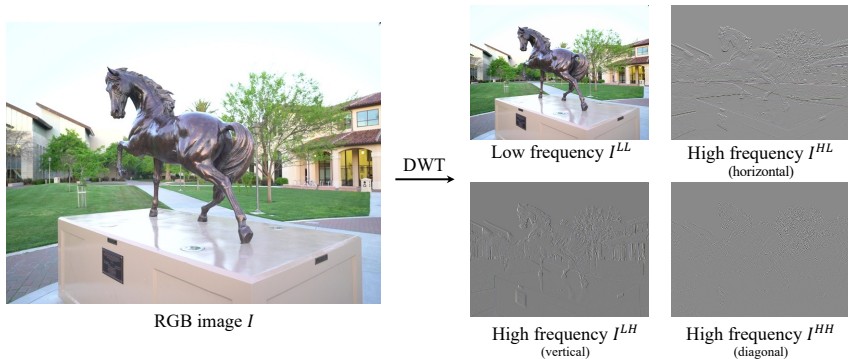

Figure 12: Illustration of the level-1 Discrete Wavalet Transformation.

In this section, we provide additional details on the Discrete Wavelet Transform (DWT) used in our Frequency-Decoupled Optimization. DWT is a widely adopted technique for frequency-domain analysis. Given an image $\mathcal{I}$, DWT decomposes it into four sub-bands: one low-frequency component and three high-frequency components corresponding to horizontal, vertical, and diagonal directions. Formally, we define the low-pass filter matrix $\boldsymbol{L}$ as:

$$L = \begin{bmatrix} \cdots & \cdots & \cdots & & & \\ \cdots & \ell_{-1} & \ell_0 & \ell_1 & \cdots & \\ \cdots & \cdots & \ell_{-1} & \ell_0 & \ell_1 & \cdots \\ & & \cdots & \cdots & \cdots & \end{bmatrix} \tag{17}$$

where $\boldsymbol{\ell}$ is the 1D low-pass wavelet filter. Similarly, the high-pass matrix $\boldsymbol{H}$ is derived from the 1D high-pass wavelet filter $\boldsymbol{h}$. We use orthogonal 1D wavelet filters such that $\boldsymbol{\ell}$ and $\boldsymbol{h}$ are the same [37, 48]. The four sub-bands are computed as:

$$\mathcal{I}^{LL} = L\mathcal{I}L^T;$$
$$\mathcal{I}^{LH} = H\mathcal{I}L^T;$$
$$\mathcal{I}^{HL} = L\mathcal{I}H^T;$$
$$\mathcal{I}^{HH} = H\mathcal{I}H^T; \tag{18}$$

We provide illustrative example in Figure 12. We extract the low-frequency feature as $\mathcal{I}_l = \mathcal{I}^{LL}$, and the high-frequency component $\mathcal{I}_h$ as the composition of directional details: $\mathcal{I}^{LH}$ (horizontal), $\mathcal{I}^{HL}$ (vertical), and $\mathcal{I}^{HH}$ (diagonal). In our implementation, we use a level-1 Haar filter for the DWT.

Table 9: Effect of 3D Gaussian percentage on reconstruction quality. PSNR denotes the peak signal-to-noise ratio at each iteration step under different initialization strategies.

| Iter. | 0 | 3000 | 6000 | 9000 | 12000 | 15000 | 18000 | 21000 | 24000 | 27000 | 30000 | PSNR |
|---|---|---|---|---|---|---|---|---|---|---|---|---|
| All 2D initialization | 0.0% | 19.0% | 29.1% | 33.7% | 35.0% | 38.9% | 39.8% | 41.4% | 43.2% | 45.7% | 47.8% | 27.25 |
| All 3D initialization | 100.0% | 86.2% | 75.3% | 69.0% | 63.9% | 59.7% | 58.7% | 58.0% | 57.4% | 57.0% | 57.0% | 27.51 |
| Random initialization | 49.9% | 58.1% | 59.9% | 57.3% | 55.1% | 52.3% | 52.5% | 52.9% | 52.9% | 53.1% | 54.2% | 27.86 |

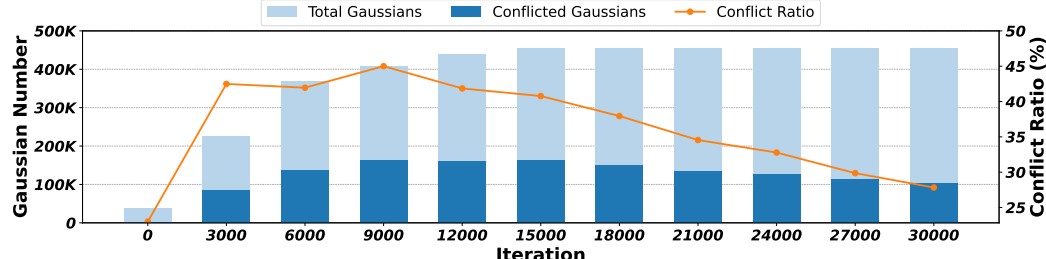

Figure 13: The number of Gaussians with conflicted gradients in different iterations during training.

## F  Gradient Conflict Analysis in Frequency-Decoupled Optimization.

We provide additional details on Frequency-Decoupled Optimization and Algorithm 1. After applying DWT, both 3D and 2D Gaussians receive gradients from the high-frequency and low-frequency losses. As discussed in Section 3.3, the distinct roles of 3D and 2D Gaussians—where 3D Gaussians prioritize fine detail and 2D Gaussians emphasize geometric structure—can lead to conflicting gradient directions.

We empirically analyze this in Figure 13, where "conflicted Gaussians" are defined as those with negative inner product between low- and high-frequency gradients, i.e., $\mathbf{g}_i^{\mathrm{low}} \cdot \mathbf{g}_i^{\mathrm{high}} < 0$ in Algorithm 1. In early training (e.g., before 15K iterations), about 45% of Gaussians exhibit such conflicts. Although the conflict ratio gradually decreases as training progresses, around 20% of Gaussians still experience conflicts at convergence. This supports our motivation that naively combining $\mathcal{L}_{\mathrm{low}}$ and $\mathcal{L}_{\mathrm{high}}$ as Eq.(8) for all Gaussians provides suboptimal supervision, and underscores the need for the proposed asymmetrical update strategy.  In Table 10, we compare different strategies for applying frequency-based supervision. As baselines, we include vanilla 3DGS and EGGS without Frequency-Decoupled Opti-

Table 10: Ablation on different use of the frequency loss. FDO stands for Frequency-Decoupled Optimization.

| Method | PSNR | SSIM | LPIPS |
|---|---|---|---|
| 3DGS | 26.12 | 0.865 | 0.099 |
| EGGS w/o FDO | 26.58 | 0.874 | 0.093 |
| EGGS w/ DWT | 26.82 | 0.877 | 0.092 |
| EGGS w/ DWT + mask | 27.07 | 0.881 | 0.089 |
| EGGS | 27.34 | 0.895 | 0.083 |

mization (EGGS w/o FDO), which includes hybrid rasterization and type exchange but no frequency regularization. EGGS w/ DWT applies frequency losses directly as in Eq.(8), yielding only marginal gains due to unresolved gradient conflicts (Figure13). A simple alternative is to mask out conflicting frequency gradients—for example, ignoring high-frequency gradients for 2D Gaussians. We denote this variant as EGGS w/ DWT + mask. While masking helps reduce conflicts, it may discard useful gradient signals. In contrast, our full method achieves the best performance by leveraging gradient projection to suppress only the conflicting components while retaining informative gradients. For theoretical background on gradient projection and conflict resolution, we refer readers to [40].

Table 11: Comparison of inference efficiency. We report the average FPS in each dataset.

| Method | LLFF | Tanks&Temples | Mip-NeRF360. |
|---|---|---|---|
| 3DGS | 323 | 158 | 145 |
| 2DGS | 187 | 59 | 76 |
| GaussianPro | 308 | 166 | 121 |
| EGGS | 268 | 125 | 104 |

## G  Inference Efficiency

As shown in Table 11, we compare the rendering efficiency of different methods in terms of frames per second (FPS). During training, the number of parameters significantly impacts performance, as backpropagation and parameter updates are computationally expensive. In contrast, inference

efficiency is primarily determined by the rasterization strategy. 3DGS-based methods, including GaussianPro, employ affine projection-based rasterization, which is efficient but less accurate, resulting in higher FPS at inference. Since both 3DGS and GaussianPro use the same projection-based rasterization pipeline, the difference in their inference speed mainly arises from model size—that is, the number of Gaussians used. While the number of primitives affects performance, its influence remains moderate given the similar scale of models.

In contrast, 2DGS adopts a ray–splat–intersection rasterization pipeline, which provides improved geometric accuracy but is more computationally intensive, resulting in slower rendering. EGGS integrates both 2DGS and 3DGS rasterization strategies in a hybrid manner, achieving a favorable balance between accuracy and efficiency. While EGGS 's FPS is slightly lower than that of 3DGS, it remains significantly faster than 2DGS. Additionally, EGGS benefits from a shorter training time than 3DGS, owing to its reduced model size and more effective optimization dynamics.

## H  Discussion

**Broader Impact.** This work introduces an exchangeable hybrid Gaussian splatting framework that improves the trade-off between geometry accuracy and appearance fidelity in neural rendering. By enabling flexible type adaptation and frequency-aware supervision, our method can enhance 3D reconstruction quality in both synthetic and real-world scenarios. Potential applications include autonomous driving, augmented reality, and robotics, where accurate scene geometry and photorealism are both essential. While our approach primarily targets academic benchmarks, it may inform future developments in real-time perception systems.

**Limitations.** The current initialization of Gaussian types is random and does not incorporate semantic or structural cues from the sparse point cloud, which may limit early-stage optimization. Incorporating semantic priors could improve convergence and final quality. Additionally, as a general-purpose representation, our method has not been explicitly tested under extreme conditions such as low-light environments, highly reflective surfaces, or scenes with significant transient content. Evaluating and adapting the framework to such challenging scenarios may further demonstrate the robustness and versatility of the hybrid representation.

