# OpenReview forum: "EGGS: Exchangeable 2D/3D Gaussian Splatting for Geometry-Appearance Balanced Novel View Synthesis"
_NeurIPS.cc/2025/Conference — NeurIPS 2025 spotlight_

### Official Review · Reviewer_wtGw · 2025-06-06

**Clarity:** 4
**Significance:** 3
**Originality:** 4
**Rating:** 5
**Confidence:** 4

**Summary:**

The EGGS proposes a novel hybrid Gaussian Splatting representation, which leverages 2D Gaussian primitives to warrant geometric accuracy while utilizing 3D Gaussian primitives to preserve details. To achieve this, the authors first develop the Hybrid Gaussian Rasterization which allows projection-based (3DGS) and ray–splat–intersection-based rasterization (2DGS) simultaneously. Then, an Adaptive Type Exchange mechanism is proposed to exchange 2DGS and 3DGS according to a predefined effective rank criterion. Finally, a Frequency-Decoupled strategy is proposed to encourage the low-frequency components to guide scene geometry and the high-frequency components to refine appearance.

**Questions:**

In Figure 11, the authors report the distribution of two Gaussian types during training. However, it will be more interesting to visualize the distribution of two Gaussian types corresponding to a specific rendering result. The rendered distribution can further verify the hypothesis that 2D Gaussians capture geometry structures while 3D Gaussians contribute to finer scene details.

**Ethical Concerns:**

["NO or VERY MINOR ethics concerns only"]

**Final Justification:**

The provided rebuttal addressed all my concerns. I believe incorporating the feedback from all reviewers can further polish the quality of this paper. I am glad to keep my original rating of 'Accept (5)'.

**Limitations:**

Yes

**Paper Formatting Concerns:**

No Formatting Concerns

**Quality:**

3

**Strengths And Weaknesses:**

The proposed EGGS is innovative and insightful. The main contribution, a Hybrid representation, may inspire many subfields in the community, such as few-shot and OOD NVS. The experiments and ablations are comprehensive. Also, the paper is well organized and easy to follow. However, I suggest the authors make more efforts to improve the supplementary video. Compared to a webpage composed of discrete results, a complete video containing all comparisons and ablations can make the results more intuitive and outstanding, since I find the results on the provided anonymous link are compressed to some extent.

---

> ### Author Rebuttal · Authors · 2025-07-31
>
> We thank Reviewer wtGw for their thorough reading of our manuscript, valuable feedback, and for recognizing the contributions of EGGS. The reviewer highlighted several positive aspects of our work: they considered our method novel and insightful, effectively combining the advantages of 3D and 2D Gaussians; found the evaluation to be comprehensive, sufficiently validating the effectiveness of our approach; and acknowledged its potential impact across multiple subfields in 3D reconstruction. Below, we address the reviewer’s concerns.
>
> **Q1: Improving the Videos and Demo**
>
> We appreciate the reviewer’s suggestion regarding improving the visualization videos and demo website. At the time of submission, we were constrained by the use of an anonymous GitHub repository, which has limited storage and does not support HTML or embedded MP4 files. As a result, we had to convert videos to lower resolution GIFs and rely on markdown for presenting results, which impacted both quality and flexibility.
>
> In the revised version, we will make efforts to enhance the demo website by including high-resolution videos across different scenes, making comparisons more accessible and intuitive.
>
> **Q2: Visualization of Gaussian type distribution in rendered images**
>
> We thank the reviewer for the careful reading of our manuscript and for recognizing our analysis of Gaussian type distribution during training. We will include additional visual examples showing the spatial distribution of 2D and 3D Gaussians in rendered images to better support our hypothesis.
>
> We sincerely appreciate the reviewer’s suggestions. Due to the rebuttal policy, we are not permitted to submit additional figures or videos or update the anonymous links at this stage. However, we will incorporate these visualizations in the revised version to better illustrate the key ideas and performance of our method.

---

> > ### Comment · Reviewer_wtGw · 2025-08-04
> > **Official Comment by Reviewer wtGw**
> >
> > Thank you for addressing all my concerns in your rebuttal. I wish you the best of luck with your research!

---

> > > ### Author Response · Authors · 2025-08-04
> > > **Thank you**
> > >
> > > We thank the reviewer for their time and careful reading of our rebuttal. We are pleased that all concerns have been addressed and sincerely appreciate the supportive feedback.

---

### Official Review · Reviewer_bqq2 · 2025-06-30

**Clarity:** 3
**Significance:** 3
**Originality:** 3
**Rating:** 5
**Confidence:** 4

**Summary:**

This paper proposes EGGS, to combine the benefits of 3D- and 2D- Gaussians. The authors developed and implemented the rasterizer, the Gaussian type change mechanism, and the optimization method for the hybrid representation. The proposed representation achieves a better trade-off between the appearance and the geometry.

**Questions:**

- What is the dataset used in Tab. 4?
- Is there more visual results regarding the reconstructed geometry (e.g. the rendered mesh, normal map video, or the depth map video)?

My main concern is about the limited evaluation of the proposed method. I would be willing to raise the rating to 5 if it is addressed.

**Ethical Concerns:**

["NO or VERY MINOR ethics concerns only"]

**Final Justification:**

Thanks to the authors' explanation. It addressed my concerns about the evaluation and the baselines. It will be appreciated if the authors could update the final revision with the comparison to the new baseline, evaluation of geometry, and the visual results, as it is stated in the rebuttal.

**Limitations:**

yes

**Quality:**

3

**Strengths And Weaknesses:**

Strengths:
- The motivation is clear and the techniques are sound.
- The paper is well-written and easy to follow.
- The proposed representation achieves state-of-the-art performance in NVS across different datasets.

Weakness:
- Insufficient and Inconsistent baselines:
    - Given HybridGS [1] is an important related work, it should serve as a baseline.
    - Why the baselines for evaluating the appearance (Tab. 1) and Geometry (Tab.2) are different, what is the standard for choosing the baselines?
- Insufficient evaluation of the reconstructed geometry:
    - The geometry (depth map) of the proposed method in Fig. 9 seems not outperform the baselines (even the 3DGS), especially in the roof region, where significant artifacts can be observed.
    - No quantitative results for geometry evaluation on the Temple and Tanks dataset.

---

> ### Author Rebuttal · Authors · 2025-07-31
>
> We thank Reviewer bqq2 for careful reading of our manuscript, insightful suggestions and recognizing positive aspects of our submission, including combining the benefits of 3D and 2D Gaussians and improving the performance across datasets. Below, we address the reviewer's concerns.
>
> **Q1: Insufficient Evaluation**
>
> **Comparison with HybridGS.** As noted in Appendix B, HybridGS did not release its training code at the time of our submission, making it difficult for a direct comparison. More importantly, the two approaches target different problem settings and propose orthogonal optimizations.
>
> - HybridGS specifically focuses on dynamic scenes with transient objects. HybridGS represents the 3D scene entirely with 3D Gaussians and leverages single-view image-space 2D Gaussians to fit transient objects. The 3D Gaussians and image-space 2D Gaussians are treated separately and are not exchangeable.
> - In contrast, EGGS focuses on general reconstruction settings and represents the 3D scene using an exchangeable hybrid of 2D and 3D Gaussians. Our hybrid representation can be integrated into the HybridGS framework by replacing its 3D Gaussians with our exchangeable hybrid representation.
>
> For completeness, we provide quantitative comparisons on two dynamic scenes of different occlusion levels from the NeRF On-the-go dataset. "3DGS" and "EGGS" denote direct reconstruction of the dynamic scene without decoupling transient objects and "+ HybridGS" means HybridGS is used to decouple the static scene and transient objects. We note that at the time of rebuttal, the authors of HybridGS had released only their trained models, not the training code. Therefore, we directly leverage the single-view image-space Gaussians from their checkpoints to remove the transients in "EGGS+HybridGS".
>
> | Medium Occlusion (Corner)  | PSNR(↑) | SSIM(↑) | LPIPS(↓) |
> |------------------|----------------|-------|--------|
> | 3DGS             | 20.90          | 0.713 | 0.241  |
> | 3DGS + HybridGS  | 25.03          | 0.847 | 0.151  |
> | EGGS             | 21.21          | 0.745 | 0.228  |
> | EGGS + HybridGS  | 25.74          | 0.896 | 0.122  |
>
> | High Occlusion (Patio-High)  | PSNR(↑) | SSIM(↑) | LPIPS(↓) |
> |------------------|--------------|-------|--------|
> | 3DGS             | 17.29        | 0.604 | 0.363  |
> | 3DGD + HybridGS  | 21.77        | 0.741 | 0.211  |
> | EGGS             | 17.56        | 0.638 | 0.352  |
> | EGGS + HybridGS  | 22.23        | 0.784 | 0.208  |
>
> As shown in the above table, directly reconstructing dynamic scenes with 3DGS leads to inferior appearance performance. This is because transient objects lead to significant inconsistency across different viewpoints. HybridGS mitigates the viewpoint discrepancy by decoupling transient objects from the 3D scene, improving the appearance by a significant margin. EGGS alone still suffers from transient objects. However, EGGS outperforms the 3DGS counterpart when transients are removed by HybridGS.
>
>
> **Standard for choosing baselines.**
> Gaussian Splatting-based methods are developed with a varying focus, some emphasizing appearance fidelity while others prioritize geometric accuracy. To provide a comprehensive evaluation, we consider both aspects. For appearance and geometry, we select state-of-the-art methods that excel in the respective aspect as baselines, allowing us to highlight the strengths of our proposed method.
>
> We note that while most Gaussian Splatting-based methods report appearance metrics such as PSNR, only a subset with an explicit focus on geometry report metrics like Chamfer Distance. In previous works, geometry-focused methods such as SuGaR, 2DGS, and GOF are commonly used as baselines for benchmarking geometry accuracy (e.g., Table 2 in [1], Table 1 in [2]). In line with this practice, we adopt these methods as baselines in Table 2 to assess the geometry performance of our approach.
>
> **Q2: Geometry Evaluation**
>
> **Qualitative comparison on Tanks&Temples.** In Figure 9, we present rendered RGB images and depth maps on the Tanks&Temples dataset. The main artifacts in our depth maps appear near the sky regions. However, depth in these areas is typically unreliable in Gaussian Splatting-based methods, and is often explicitly masked out during evaluation (e.g., Figure 7 in [3]). In our current visualization, we did not apply such masking, which likely exaggerates the perceived artifacts. We believe the inaccuracies near the sky do not meaningfully reflect overall depth quality. While not as clean as 2DGS, our method produces more accurate depth maps than 3DGS for foreground objects and buildings.
>
> We provide a quantitative comparison below to further verify the geometry accuracy. We will also update the visualizations and include additional depth map videos in the revised version to improve clarity.
>
> **Quantitative comparison on Tanks&Temples.**
> We provide a quantitative geometric comparison on the Tanks&Temples dataset. Following prior works, we report the F1 score based on alignment between the predicted and ground-truth point clouds.
>
> | F1 score(↑) | Barn | Caterpillar | Courthouse | Ignatius | Meetingroom | Truck | Avg. |
> |--------|------|-------------|------------|----------|--------------|--------|------|
> | 3DGS   | 0.13 | 0.08        | 0.09       | 0.04     | 0.01         | 0.19   | 0.09 |
> | SuGaR  | 0.14 | 0.16        | 0.08       | 0.33     | 0.15         | 0.26   | 0.19 |
> | 2DGS   | 0.41 | 0.23        | 0.16       | 0.52     | 0.17         | 0.45   | 0.32 |
> | GOF    | 0.51 | 0.41        | 0.28       | 0.68     | 0.28        | 0.58   | 0.46 |
> | Ours   | 0.35 | 0.18        | 0.12       | 0.45     | 0.15         | 0.41   | 0.28 |
>
> As shown in the above table, our method consistently outperforms 3DGS in F1 score across all scenes. While 2DGS has a slightly higher F1 score, 2DGS has inferior appearance performance, as shown in Table 1 in our manuscript. While geometry-focused methods like 2DGS and GOF prioritize geometry accuracy for surface reconstruction tasks, they often lead to decreased appearance fidelity even compared to the 3DGS baseline. This aligns with our analysis on the DTU dataset in Table 2. In contrast, our method achieves a better balance, improving the appearance while maintaining high geometry accuracy.
>
> **Q3: Dataset used in Table 4**
>
> The ablation study in Table 4 is conducted on the LLFF dataset. We will clarify this in the revised version.
>
> **Q4: More Visualization**
>
> Due to the rebuttal policy, we are not permitted to submit additional figures or videos at this stage. However, in addition to the quantitative comparisons already provided in our manuscript and in the response to Q2, we will make sure to include more qualitative visualizations in the revised version and on the demo website. These will include depth map videos, normal map videos, and rendered meshes.
>
> We sincerely appreciate the reviewer’s constructive feedback. In the revised version, we will include the comparison with HybridGS and additional geometry evaluation on Tanks&Temples. We will also provide more visual results to highlight the reconstructed geometry. We hope our response addresses the reviewer’s concerns, and we would be grateful for a positive reconsideration of the score.
>
> [1] Huang, Binbin, et al. "2D Gaussian Splatting for Geometrically Accurate Radiance Fields."
>
> [2] Hyung, Junha, et al. "Effective Rank Analysis and Regularization for Enhanced 3D Gaussian Splatting."
>
> [3] Cheng, Kai, et al. "GaussianPro: 3D Gaussian Splatting with Progressive Propagation."

---

> > ### Comment · Reviewer_bqq2 · 2025-08-03
> >
> > Thank you for your reply. It addressed my concerns.

---

> > > ### Author Response · Authors · 2025-08-04
> > > **Thank you**
> > >
> > > We thank the reviewer for carefully reading our rebuttal and for the positive feedback. We are pleased that our response has addressed the reviewer's concerns.

---

### Official Review · Reviewer_2p4H · 2025-07-03

**Clarity:** 3
**Significance:** 3
**Originality:** 4
**Rating:** 5
**Confidence:** 4

**Summary:**

The paper propose a new Gaussian Splatting framework with the aim of combining the 3D reconstruction capabilities of 2D gaussians with the rendering quality of a 3D gaussian representation. For this the paper introduces a hybrid representation which let's gaussians adaptively switch between 3D and 2D, depending on which capabilities are needed at this point of the scene. Supervision is done in the frequency domain, which allows to split the scene content into general geometry (low frequency) and finer details relevant for rendering (high-frequency).

**Questions:**

1. What is the purpose of $\lambda_z$ in Eq. 6? If it is just a scaling factor, couldn't the same role not also be fulfilled by the parameters of the gating function in Eq. 7?
2. Can you give a bit more intuition into the scale modulation (L196)? From my understanding it just seems like an engineering trick to overcome visual artifact's on transparent areas.

I am more than open to increase the score should the authors address the lacking baselines and provide information on the rendering speed.

**Ethical Concerns:**

["NO or VERY MINOR ethics concerns only"]

**Final Justification:**

The rebuttal submitted by the authors managed to address of my concerns (W1-W4) and answered my questions regarding the scale modulation (Q1, Q2). Therefore I decided to raise the score from "Borderline Accept" to "Accept".

**Limitations:**

Yes

**Paper Formatting Concerns:**

Fig 5. and Fig 6 have really tight bounds which makes it hard to differentiate from surrounding text.

**Quality:**

3

**Strengths And Weaknesses:**

**Strengths**

- **1. Performance:** The method has SoTA rendering performance both in qualitative and quantitative evaluations on common datasets (Mip-NeRF360, LLFF, Tanks&Temples). Additionally, the authors demonstrate competitive geometry reconstruction capabilities fulfilling the claim of balancing appearance and geometry.
- **2. Idea and novelty:** The idea of combining the best of both worlds, rendering quality of 3D Gaussians and geometric reconstruction capabilities of 2D Gaussians is novel and well-motivated by the authors. Separating the scene into high and low-frequency content is a good idea, and  adaptive approach of separating the scene into high and low frequency components is a good idea which seems to work well in practice.

**Weaknesses**

- **1. Lacking SoTA baselines**: The evaluation lacks a comparison with the SoTA method 3DGS-MCMC (Kheradmand et al. NeurIPS 2024), although shortly mentioning it in the appendix (L580). Additionally the  evaluation should also include NeRF-based methods for completeness, for which I would think an inclusion of MipNeRF360 (Barron et al. CVPR 2022) would be sufficient.
- **2. Information about rendering speed:** The evaluation does not contain any information about the actual rendering speed of the method and how it compares to the baselines. I would expect this as additional column in Tab. 1 or Tab. 3.
- **3. Ablation study lacks geometric metrics:** The main claim of the paper is to balance rendering and reconstruction quality, therefore the ablation study (Tab. 4) should also contain information on how the model components influence the geometric reconstruction.
- **4. Related work too focused on rendering:** The related work focuses purely on rendering-based methods but should at least acknowledge recent methods purely reconstruction such as DUSt3R (Wang et al. CVPR 2024).

---

> ### Author Rebuttal · Authors · 2025-07-31
>
> We thank Reviewer 2p4H for their thorough reading of our manuscript, constructive feedbacks and recognizing several positive aspects of our submission, including the novelty of our idea and the improved performance of EGGS in both appearance and geometry. Below, we address the main concerns raised in the review.
>
> **Q1: Comparison with SoTA baselines**
>
> We thank the reviewer for acknowledging that EGGS improves reconstruction quality and achieves a better balance between appearance and geometry. We provide quantitative comparisons with two additional strong baselines: the SoTA 3DGS-based method 3DGS-MCMC and the NeRF-based method Mip-NeRF360. We evaluated the performance on the Mip-NeRF 360 and LLFF dataset. We also incorporate 3DGS-MCMC into our framework by replacing the vanilla 3DGS component.
>
> | Mip-NeRF 360 Dataset                 | PSNR(↑) | SSIM(↑) | LPIPS(↓) |
> |------------------------|------|------|-------|
> | Mip-NeRF 360           | 27.88 | 0.835  |   0.201    |
> | 3DGS                   | 27.43 | 0.814  |   0.257    |
> | 3DGS-MCMC              | 28.11 | 0.867  |  0.188    |
> | Ours                   | 27.96 | 0.851  |  0.192     |
> | Ours + 3DGS-MCMC       | 28.58    |  0.892    |  0.184  |
>
> | LLFF Dataset               | PSNR(↑) | SSIM(↑) | LPIPS(↓) |
> |------------------------|------|------|-------|
> | Mip-NeRF 360           | 26.79 | 0.871 |  0.092     |
> | 3DGS                   | 26.12 | 0.865 |  0.099   |
> | 3DGS-MCMC              | 27.81    |  0.905 |  0.077 |
> | Ours                   | 27.34 | 0.895 |  0.083   |
> | Ours + 3DGS-MCMC       | 28.03 | 0.931 |  0.069   |
>
> As shown above, our method achieves comparable performance with both 3DGS-based and NeRF-based SoTA baselines. More importantly, by replacing vanilla 3DGS with 3DGS-MCMC (i.e., Ours + 3DGS-MCMC), our method can outperform the SoTA baselines. This suggests that while current EGGS is built on the vanilla 2DGS and 3DGS, more advanced variants can be incorporated to further improve the performance.
>
> **Q2: Rendering Speed**
>
> We provided an analysis of the inference speed in terms of frames per second (FPS) in Table 10 of the supplementary materials (included in the .zip file). For convenience, we provide the results below.
>
> | FPS(↑)       | LLFF | Tanks&Temples | Mip-NeRF 360 |
> |--------------|------|----------------|--------------|
> | 3DGS         |  323    |   158            |    145           |
> | 2DGS         |  187    |   59             |    76            |
> | GaussianPro  |  308    |   166             |   121           |
> | Ours         |  268    |   125             |   104           |
>
> The rendering speed is mostly determined by the rasterization technique. While 2DGS typically has the smallest model size, as shown in Table 3 in our manuscript, its inference speed is slower than 3DGS-based methods. This is because the ray-splat intersection used in 2DGS incurs more intensive computation than affine approximation-based rasterization in 3DGS. Our method combines the two rasterization techniques and has a higher FPS than 2DGS. We note that while EGGS has a lower FPS than pure 3DGS-based method, our method achieves a better balance between model size, training and inference efficiency and reconstruction quality. We will add an additional column to Table 3 to clearly indicate the inference efficiency of each method in our revised version.
>
> **Q3: Ablation on Geometry**
>
> In Table 4, we perform the ablation study on LLFF dataset. Since LLFF dataset does not provide ground-truth point clouds or depth maps, we did not report geometry metrics in that table. We provide additional ablation results on the DTU and Tanks&Temples dataset, which include ground-truth point cloud. Following the common practice in geometry accuracy evaluation, we report Chamfer Distance (CD) on DTU and F1 score on Tanks&Temples.
>
> | ID | Repr. | Hyb. | Ex. | Freq. | CD(↓)  | F1(↑)  |
> |----|-------|------|-----|-------|-----|-----|
> | i  |   3D         |  ✖    |  ✖   |   ✖    |  1.96    | 0.09    |
> | ii  |   3D+2D     |  ✖    |  ✖   |   ✖    |  2.01    | 0.08    |
> | iii  |   3D+2D    |  ✔    |  ✖   |   ✖    |  1.31   |  0.16   |
> | iv  |  3D+2D      |  ✔    |  ✔   |   ✖    |  1.07   |  0.21   |
> | v  |   3D         |  ✖    |  ✖   |   ✔    |  1.95   |  0.09   |
> | vi  |  3D+2D      |  ✔    |  ✖   |   ✔    |  1.18   |  0.18   |
> | vii  | 3D+2D      |  ✔    |  ✔   |   ✔    |  0.91    | 0.28 |
>
> As shown in the above table, merely mixing 2D and 3D Gaussians (row ii) or directly adding frequency regulation (row v) can hardly improve the geometry accuracy. Our hybrid rasterizer (row iii) improves both CD and F1 score by a significant margin. This indicates the importance of the ray-splat intersection based rasterization. Incorporating adaptive type exchange (row iv) and frequency decoupled optimization (row vii) allows a more flexible representation, more effectively exploiting the geometric accurateness of 2D Gaussians.
>
> **Q4: Scope of related works**
>
> We appreciate the reviewer’s suggestion to broaden the related work section. Purely reconstruction-focused methods such as DUSt3R are indeed valuable, particularly in settings where camera poses are not available. In this work, we adopt a commonly used setting for rendering-based methods, where camera poses are assumed to be known. In practice, such poses are often obtained through tools like COLMAP or more recent methods like DUSt3R[1] and VGGT[2].
>
> In the revised version, we will expand the related work section to acknowledge and discuss recent purely reconstruction methods in more detail.
>
> **Q5: The functionality of $\lambda_z$.**
>
> Our scale modulation consists of two components: Eq.(7), which applies a soft gating to the raw $z$-scale $s_z$, and Eq.(6), which modulates the opacity using the activated $z$-scale $s_{z*}$. Each plays a distinct role.
>
> - Eq.(7) controls *when* the $z$-scale becomes active by smoothly mapping $s_{z}$ to $s_{z*}$ using a sigmoid gate. When $s_z$ is negligibly small, the gating suppresses the activation and $s_z*$ approaches zero. In this case, the Gaussian behaves like a 2D primitive. As $s_z$ increases beyond the threshold, $s_z*$ effectively approaches the original scale $s_z$.
> - Eq.(6), on the other hand, defines *how much* the activated $z$-scale influences opacity. The parameter $\lambda_z$ controls the strength of this correlation: as $s_{z*}$ increases, the opacity decreases smoothly, regulated by $\lambda_z$. This formulation enables opacity to act as a proxy signal to optimize $s_z$ for 2D Gaussians, which otherwise do not receive gradient updates to their $z$-scale.
>
> In summary, Eq.(7) controls the activation of the $z$-scale, while Eq.(6) introduces a soft correlation between scale and opacity to facilitate learnable type transitions. Although it is possible to combine them into a single equation, we separate them for clarity in both concept and implementation.
>
> **Q6: Clarification on scale modulation**
>
> In vanilla 2DGS, the $z$-scale is not involved during optimization. The primary purpose of our scale modulation is to enable 2D Gaussians to transition into 3D Gaussians when necessary. Our intuition is that this transition typically occurs when a Gaussian needs to develop volumetric capacity or represent semi-transparent regions. To support this, we introduce a correlation between $z$-scale and opacity for 2D Gaussians, which allows the $z$-scale to receive meaningful gradient updates. We believe that scale modulation is essential for enabling dynamic type exchange between 2D and 3D Gaussians, and for supporting a more flexible representation, instead of merely an engineering trick.
>
> **Q7: Figure margin**
>
> We appreciate the reviewer’s suggestion. We will increase the margins around Figure 5 and Figure 6 to improve readability.
>
> We sincerely appreciate the reviewer’s constructive suggestions. In the revised version, we will incorporate additional evaluations with SoTA methods, include efficiency and geometry metrics, expand the discussion on purely reconstruction-based methods, and clarify the scale modulation design. We hope these additions and clarifications address the reviewer’s concerns and would be grateful for a positive reconsideration of the score.
>
> [1] Wang, Shuzhe, et al. "DUSt3R: Geometric 3D Vision Made Easy."
>
> [2] Wang, Jianyuan, et al. "VGGT: Visual Geometry Grounded Transformer."

---

> > ### Comment · Reviewer_2p4H · 2025-08-05
> >
> > I thank the authors for their extensive reply to my questions and concerns! I am happy to see the inclusion of 3DGS-MCMC  improved the method performance even further. Due to the clarity of the answers (especially regarding the scale modulation and $\lambda_z$) I have no further questions.

---

> > > ### Author Response · Authors · 2025-08-05
> > > **Thank you**
> > >
> > > We thank the reviewer for taking the time to read our rebuttal and for the positive feedback. We are pleased that the reviewer's concerns have been addressed.

---

### Official Review · Reviewer_5biL · 2025-07-07

**Clarity:** 3
**Significance:** 3
**Originality:** 3
**Rating:** 5
**Confidence:** 4

**Summary:**

This paper proposes EGGS, Exchangeable 2D/3D Gaussian Splatting for Geometry-Appearance Balanced Novel View Synthesis. 3D Gaussian Splatting enables real-time rendering with high appearance fidelity, but suffers from multi-view inconsistency. On the other hand, 2D Gaussian Splatting enforces multi-view consistency but compromises texture details. EGGS aims to meet a middle ground to get the best of both worlds: appearance fidelity  as well as multi view consistency.

To enable this, a few technical novelties are needed: Hybrid Gaussian Rasterization, a unified rendering framework that supports both 2DGS and 3DGS rendering, Adaptive Type Exchange, which enables an exchangeable hybrid of 2D and 3D Gaussians, and Frequency-Decoupled Optimization, a regularization strategy in the frequency domain that allows for asymmetric optimization of 3DGS and 2DGS based on their individual traits. First, each Gaussian is augmented by a ‘type’ variable, that codifies if it is a 2DGS or 3DGS representation. Basically, a 2DGS has a z scale equal to 0. Then, the contributions of each Gaussian is calculated based on its projected distance from the pixel and are together rendered. Additionally,  Gaussians can move from one type to another, based on their effective rank (number of dimensions). However, since 2DGS itself does not backpropagate gradients to the z scale, to enable Gaussians to learn from z scale, opacity is defined to be an inverse function of z scale. Finally, wavelet based decomposition allows for each type of Gaussian to be supervised on parts that they are stronger on, enabling better representation.

Across several datasets, and when compared with a range of baseline methods, EGGS shows improved qualitative and quantitative performance. The hybrid representation also allows for efficient use of Gaussians, and fast training and rendering. Finally, ablation analyses indicate the benefit of each component.

**Questions:**

Questions:
1. Is there any empirical or mathematical evidence that can further emphasize the relationship between opacity and z scale assumed in this work?
2. If a Gaussian is incorrectly initialized as 2D in a region that needs 3D (or vice versa) do we have any analysis to show that the Gaussians are highly likely to recover to the correct Gaussian type?
3. Would it be possible to have a primer section on the Math of 2DGS and 3DGS and how they relate more explicitly at the beginning of Sec 3, to make readability better?

**Ethical Concerns:**

["NO or VERY MINOR ethics concerns only"]

**Final Justification:**

I thank the authors for their responses to the questions. I am happy to maintain the Accept rating for this paper, and hope the authors incorporate the feedback from all reviewers in the revision.

**Limitations:**

Yes, in the appendix. It would be good if this can be in the main paper.

**Quality:**

3

**Strengths And Weaknesses:**

Strengths:

1. The paper is well written, easy to follow,
2. The method is well motivated, explained well, and takes into account various factors that are necessary.
3. Results show superior qualitative and quantitative performance across a range of baselines, and motivate the various method components well through ablation analyses.

Weaknesses:
1. I am a bit unsure about the inverse relationship between opacity and z scale. Complex non-planar yet solid structures may benefit from being 3D but opaque, so not sure if this is a mathematically grounded assumption.
2. In terms of the initialization of the Gaussians, it seems the initialization is random for a Gaussian to be either 2D or 3DGS. But this seems quite non-optimal: why can we not use the wavelet-based decomposition and some coarse metric of scene texture to better blend Gaussian initializations?

---

> ### Author Rebuttal · Authors · 2025-07-31
>
> We thank Reviewer 5biL for their constructive feedback and for recognizing the contributions of our work, including the soundness of our methods and superior qualitative and quantitative performance. Below, we address the main concerns raised in the review.
>
> **Q1: The relationship between opacity and z-scale**
>
> **Representing non-planar yet solid structures.** We appreciate the reviewer’s insight regarding the potential importance of 3D but opaque Gaussian primitives for representing non-planar yet solid structures. This observation actually aligns with our design: the scale modulation between opacity and z-scale applies **only** to 2D Gaussians. Non-planar yet solid structures can still be effectively captured by 3D Gaussians, which remain unaffected by this modulation. During training, the model can either densify 3D Gaussians or convert suitable 2D Gaussians into 3D ones to better represent such regions.
>
> **Empirical analysis on the relationship of opacity and z-scale.** The primary motivation for modulating the z-scale of 2D Gaussians is to enable their transition into 3D Gaussians when necessary. Our intuition is that such a transformation typically occurs when the model needs to acquire volumetric capacity or represent semi-transparent regions. To facilitate this, we introduce an inverse relationship between opacity and z-scale only during training.
>
> To verify the rationale behind scale modulation, we conducted training *without* explicit z-scale modulation. In this setup, we employed affine approximation-based rasterization for 2D Gaussians to enable gradient flow to the z-scale. While this rasterization is not geometrically accurate, it suffices to observe the dynamics between opacity and z-scale during optimization. We analyze the cosine similarity between the gradient of opacity ($\nabla_{\alpha}$) and z-scale ($\nabla_{s_z}$) for 2D Gaussians.
>
> | Iteration | 3000   | 6000   | 9000   | 12000  | 15000  | 18000  | 21000  | 24000  | 27000  | 30000  |
> |-----------|--------|--------|--------|--------|--------|--------|--------|--------|--------|--------|
> | cos($\nabla_{\alpha}$, $\nabla_{s_z}$)     | -0.221 | -0.216 | -0.244 | -0.241 | -0.253 | -0.245 | -0.201 | -0.214 | -0.248 | -0.196 |
>
>
> As shown in the table, $\nabla_{\alpha}$ and $\nabla_{s_z}$ have negative cosine similarity across different training iterations. This suggests 2D Gaussians with increasing z-scale tend to exhibit decreasing opacity. Such behavior supports our design of soft modulation and validates the effectiveness of our design.
>
>
> **Q2: Alternative initialization of Gaussian type**
>
> While more advanced initialization strategies may improve reconstruction quality, designing such initialization is non-trivial. For instance, wavelet-based decomposition can effectively decouple frequency information in the 2D image space. However, during initialization, 2D pixels do not directly correspond to 3D points, which makes it challenging to apply Discrete Wavelet Transform for initializing Gaussian type.
>
> In our response to Q3 below, we show that different initializations often converge to Gaussian models with similar type distributions and comparable appearance performance. This empirical observation suggests that random initialization serves as a reasonable and effective starting point. As mentioned in our limitations section, 3D point segmentation could potentially enhance initialization. For example, we can segment the initial 3D points into distinct regions and assign different Gaussian types to different semantic segments. This might lead to improved initialization and more efficient training. However, determining an optimal mapping between semantic labels and Gaussian types remains challenging. We thus leave the development of more advanced initialization techniques to future work.
>
> **Q3: Can 2D Gaussians change to 3D if initialization is incorrect?**
>
> A key feature of EGGS is its exchangeable representation, which allows each Gaussian primitive to change its type as needed, regardless of its initial type. To verify this capability, we conduct a simple experiment by investigating three initialization scenarios: we initialize all Gaussians as 2D, all as 3D, or with random type assignments, and observe the distribution of Gaussian types throughout training. Below, we show the percentage of 3D Gaussians at different iterations.
>
> | Percentage of 3D Gaussians  | 0      | 3000   | 6000   | 9000   | 12000  | 15000  | 18000  | 21000  | 24000  | 27000  | 30000  | PSNR  |
> |--------|--------|--------|--------|--------|--------|--------|--------|--------|--------|--------|--------|--------|
> | All 2D initialization | 0.0%   | 19.0%  | 29.1%  | 33.7%  | 35.0%  | 38.9%  | 39.8%  | 41.4%  | 43.2%  | 45.7%  | 47.8%  | 27.25 |
> | All 3D initialization | 100.0% | 86.2%  | 75.3%  | 69.0%  | 63.9%  | 59.7%  | 58.7%  | 58.0%  | 57.4%  | 57.0%  | 57.0%  | 27.51 |
> | Random initialization | 49.9%  | 58.1%  | 59.9%  | 57.3%  | 55.1%  | 52.3%  | 52.5%  | 52.9%  | 52.9%  | 53.1%  | 54.2%  | 27.86 |
>
> As shown in the above table, even when the model is initialized entirely with 2D Gaussians, part of the Gaussians are converted to 3D Gaussians during training, leading to a hybrid model in the final stage. This demonstrates that 2D Gaussians can indeed transition to 3D types during training, despite potentially incorrect initialization, and vice versa.
>
> **Q4: Improve readability**
>
> Due to space constraints, we included an introduction to the mathematical formulation of 2DGS and 3DGS rasterization in Appendix B. In the revised version, we will add a brief preliminary section at the beginning of Section 3 to further improve readability.
>
> We sincerely appreciate the reviewer’s constructive suggestions. We will include detailed discussions on the scale modulation and initialization in our revised version to offer more insights into our method.

---

> > ### Comment · Reviewer_5biL · 2025-08-03
> > **Thank you for your response**
> >
> > Thank you very much for your response to my questions. My questions have been addressed through the rebuttal.

---

> > > ### Author Response · Authors · 2025-08-04
> > > **Thank you**
> > >
> > > We thank the reviewer for their time, thoughtful feedback, and careful reading of our rebuttal. We are pleased that all concerns have been addressed in the discussion.

---

### Note · Authors · 2025-08-12

We thank all reviewers for their time and constructive feedback, and we appreciate their recognition of the positive aspects of our work, including:

- **Balanced appearance and geometry.** All reviewers noted that our method effectively balances appearance and geometry, achieving superior rendering performance while maintaining competitive geometric accuracy.
- **Novel and sound techniques.** Reviewers 5biL, 2p4H, and wtGw found our method novel, supporting an exchangeable and flexible representation of 2D and 3D Gaussians. Reviewer bqq2 found that our techniques are well-motivated and sound.
- **Comprehensive evaluation.** Reviewer 5biL and wtGw considered our evaluation comprehensive and agreed that our ablation studies validate the effectiveness of each component, including Hybrid Gaussian Rasterization, Adaptive Type Exchange, and Frequency-Decoupled Optimization.
- **Superior qualitative and quantitative performance.** All reviewers agreed that our method achieves superior qualitative and quantitative results across multiple datasets, outperforming a range of baselines in both appearance and geometry.
- **Clear and well-organized presentation.** Reviewers 5biL, bqq2, and wtGw found our manuscript well written and our techniques well motivated, with clear explanations.
- **Benefits to subfields.** Reviewer wtGw highlighted that our work could impact multiple subfields, including few-shot and out-of-distribution (OOD) novel view synthesis.

During the discussion phase, we provided additional analysis on initialization and scale modulation, as well as further quantitative evaluation of geometry accuracy. We also incorporated comparisons with more baselines in different settings, such as 3DGS-MCMC and HybridGS. Reviewer 2p4H found our responses informative and clear. All reviewers agreed that their concerns were addressed. We will incorporate these discussions into the camera-ready version.

We are pleased that the reviewers found our work innovative and insightful, with solid evaluation and superior performance, and that they recognized its contributions to the community.

---

### Decision · Program_Chairs · 2025-09-17

**Decision:**

Accept (spotlight)

**Comment:**

This paper proposes a 2D-3D hybrid representation for Gaussian splatting. The hybrid representation incorporates the best of both worlds (i.e., appearance and geometry). All reviewers spoke very highly of the paper (i.e., all 5 ratings after the discussion). All reviewers have agreed that the proposed method is novel and well-designed, with many potential applications in the field, and the results are excellent. The frequency-decoupled optimization was also deemed effective. The weakness questions were mostly focused on some detailed technical questions and some requests for additional information (e.g., comparison to some missing SOTA methods, some missing metrics in experiments, and some missing related works). The rebuttal addressed all these concerns, which the reviewers found satisfactory. AC agrees on the positive ratings and the novelty of the hybrid representation. Accordingly, an accept decision is recommended under the condition that the additional information provided in the rebuttals is included in the final version.